# Statistical downscaling of water vapour satellite measurements from profiles of tropical ice clouds

Giulia Carella[12], Mathieu Vrac[1], Hélène Brogniez[2], Pascal Yiou[1], and Hélène Chepfer[3]

[1]Laboratoire des Sciences du Climat et de l'Environnement (LSCE/IPSL, CNRS - CEA - UVSQ - Université Paris-Saclay), Orme des Merisiers, Gif-sur-Yvette, France
[2]Laboratoire Atmosphères, Milieux, Observations Spatiales (LATMOS/IPSL, UVSQ Université Paris-Saclay, Sorbonne Université, CNRS), Guyancourt, France
[3]Laboratoire de Météorologie Dynamique (LMD/IPSL, Sorbonne Université, Ecole Polytechnique, CNRS), Paris, France

**Correspondence:** Hélène Brogniez (helene.brogniez@latmos.ipsl.fr)

**Abstract.** Multi-scale interactions between the main players of the atmospheric water cycle are poorly understood, even in present-day climate, and represent one of the main sources of uncertainty among future climate projections. Here, we present a method to downscale observations of relative humidity available from the passive microwave sounder SAPHIR at a nominal horizontal resolution of $10 \, \text{km}$ to the finer resolution of $90 \, \text{m}$ using scattering ratio profiles from the lidar CALIPSO. With the scattering ratio profiles as covariates, an iterative approach applied to a non-parametric regression model based on Quantile Random Forest is used. This allows to effectively incorporate into the predicted relative humidity structure the high-resolution variability from cloud profiles. The finer-scale water vapour structure is hereby deduced from the indirect physical correlation between relative humidity and the lidar observations. Results are presented for tropical ice clouds over the ocean: based on the coefficient of determination (with respect to the observed relative humidity) and the Continuous Rank Probability Skill Score (with respect to the climatology), we conclude that we are able to successfully predict, at the resolution of cloud measurements, the relative humidity along the whole troposphere, yet ensuring the best possible coherence with the values observed by SAPHIR. By providing a method to generate pseudo-observations of relative humidity (at high spatial resolution) from simultaneous co-located cloud profiles, this work will help revisiting some of the current key barriers in atmospheric science. A sample dataset of simultaneous co-located scattering ratio profiles of tropical ice clouds and observations of relative humidity downscaled at the resolution of cloud measurements is available at http://dx.doi.org/10.14768/20181022001.1 (Carella et al., 2019).

## 1 Introduction

The atmospheric water cycle consists of complex processes covering a wide range of scales. At small scales, the components of the atmospheric water cycle - water vapour, clouds, precipitation (rain and snow), aerosols - interact amongst each other and with their surrounding environment through micro-physical, radiative and thermo-dynamical processes. At global scales, the atmospheric water cycle interplays with the global atmospheric circulation and the Earth radiative balance. These complex multi-scale interactions are not well understood and how the global atmospheric water cycle works in present-day climate is

the subject of intense research, e.g. within the World Climate Research Program (WCRP) core project "Global Earth Water cycle Exchanges" (GEWEX, http://www.gewex.org/) and within the WCRP grand challenge on "cloud, circulation and climate sensitivity" (https://www.wcrp-climate.org/grand-challenges). Given this poor understanding, it is challenging to anticipate how the atmospheric water cycle will evolve in the future as climate warms (Boucher et al., 2013).

A symptomatic example of this lack of knowledge is the difficulty of state-of-the-art climate models to reproduce the observed clouds and precipitation in present-day climate (Nam et al., 2012), Cesana and Chepfer (2012), Zhang et al. (2005), Kay et al. (2016), Klein et al. (2017), Lacour et al. (2017)). One of the reasons is that small-scale processes act at space and time scales smaller than the model grid-box and smaller than the model time step, therefore those processes are not represented explicitly in climate models. As a consequence, on a longer term (hundred years), the projections on how clouds

and precipitation will evolve in the future differ amongst models (Vial et al., 2016). Observations collected by field experiments and ground-based sites have provided essential knowledge on how the atmospheric water cycle works at small scale (<100 m) (Campbell et al., 2002; Intrieri et al., 2002; Shupe et al., 2006; Long et al., 2009; Wild, 2009; Manara et al., 2016), but these observations are sparse and limited in space. Thanks to their global cover and their long life-time, satellites have observed the water cycle components on a global scale for over 25 years (Gruber and Levizzani, 2008; Raschke et al., 2012; Stubenrauch et

al., 2013). However, these satellites lack some essential capabilities, such as documenting the detailed vertical structure of the water cycle components. Since 2006, the space lidar CALIPSO (Winker et al., 2017) and the space radar CloudSat (Stephens et al., 2008, 2018) provide a more detailed view of aerosols, clouds, and precipitation (light rain and snow), on a global scale. These active sensors provide new surface-blind detailed vertical profiles of aerosols (Liu et al., 2009; Sekiyama et al., 2010), clouds (Mace et al., 2009; Vaughan et al., 2009; Chepfer et al., 2010), snow precipitation (Palerme et al., 2014), Arctic

atmosphere (Kay et al., 2008; Cesana and Chepfer, 2012), light rain precipitation (Lebsock and L'Ecuyer, 2011), atmospheric heating rate profiles and surface radiation (Kato et al., 2011; Stephens et al., 2012).

    Similarly, atmospheric reanalyses, although suited for the study of integrated contents of water vapour (Obligis et al., 2009; Schröder et al., 2017), exhibit noticeable biases in the tropical water and energy budget on the vertical. As suggested by comparisons between satellite observations of single-layer upper tropospheric humidity and atmospheric reanalyses (Chuang

et al., 2010; Chiodo and Haimberger, 2010), reanalyses fail to reproduce the observed vertical correlation structure between the various layers of relative humidity in the upper troposphere, where moisture is mainly influenced by the shape of the convective detrainment profile in deep convective clouds (Folkins et al., 2002), together with drying effects induced by mixing or air intrusion from the subtropics (Pierrehumbert, 1998; Brogniez et al., 2009). On the other hand, since 2011, the passive microwave sensor "Sondeur Atmosphérique du Profil d'Humidité Intertropical par Radiométrie" (SAPHIR), provides over the

entire tropical belt (30°S - 30°N) observations of water vapour even in the presence of (non-precipitating) clouds, which are largely transparent at frequencies above 100 GHz (Brogniez et al., 2015). These detailed profiles are observed all over the Tropics, and thus are good candidates to help improving our current understanding on how the atmospheric water cycle works.

    However, if the new generation of cloud observations from space has the relevant spatial resolution (60 m on the vertically, 333 m horizontally, Chepfer et al. (2010)) and the global cover to document processes over the entire Earth, the water vapour

observations do not. The water vapour measured by SAPHIR is observed at larger spatial resolutions (with a footprint size at

nadir of 10 km) which implies that small scale horizontal heterogeneities will be missed, critical for understanding the full water cycle processes. To better understand the atmospheric water cycle, and the multi-scales interplays, it is thus of strong interest to build a pseudo-observations dataset that contains, over the entire tropical belt and during several years, simultaneous co-located profiles of water vapour and clouds at high spatial resolution relevant to process studies (480 m vertically and 330 m horizontally, Chepfer et al. (2010)). It is the purpose of this paper to build such a pseudo-observation dataset.

When combining measurements from different platforms care must be taken to account for the different spatial resolutions of the instruments (Atkinson , 2013). For spaceborne instruments, the horizontal spatial resolution or support is determined by the sensor's instantaneous field of view and is approximately equal to the size of a pixel in an image provided by that sensor. Although ideally we would like all spaceborne measurements to have the finest possible horizontal spatial resolution, in practice there is a limit imposed by the trade-off between spatial resolution, revisit time and spatial coverage: on the one hand, CALIPSO and CloudSat provide images with a fine horizontal spatial resolution (see section 2.2) but have a sparse coverage and a long revisit time due to their polar orbiting; on the other hand, SAPHIR, owing to the low inclination of its orbit, is characterized by a much higher revisit frequency and a more complete coverage, but has a lower horizontal spatial resolution (see section 2.1). The support therefore provides a limit on what a spaceborne sensor can retrieve and effectively acts as 'filter on reality' (Atkinson , 2013): different instruments with different supports will indeed view the Earth differently.

Statistical downscaling methods (Bierkens et al., 2000; Vaittinada Ayar et al., 2015) involve reconstructing a coarse-scale measured variable at a finer resolution based on statistical relationships between large- and local-scale variables. Although the typical application for these methods is to derive sub-grid scale climate estimates from GCMs outputs or reanalysis data to drive impact studies (Gutierrez et al., 2018), recent studies have started adopting the standard downscaling techniques to enhance the resolution of satellite images using available covariate data at a finer resolution (Liu and Pu, 2008; Malone et al., 2012). Following the approach taken in these studies, here we are interested in modelling, at the finer scale of the clouds measurements, the statistical relationship between the water vapour layered-vertical structure associated to ice clouds in the tropical belt and the vertical profiles of clouds provided by CALIPSO. The method employed in this study provides a general framework to effectively perform a downscaling of SAPHIR observations of relative humidity and, for unsampled locations and times, to predict the (downscaled) RH layered profiles using cloud profiles only. The main interest of this study is to test a statistical approach to overcome the barrier of the coarse footprint size of the radiometer, which implies that small scale heterogeneities in the $RH$ field are missed. The coarse vertical resolution is also critical, especially in cases where there are strong vertical gradients of moisture. For instance, at the top of the atmospheric boundary layer over the oceans in regions of shallow clouds (stratocumulus or cumulus) the boundary layer can be really moist, near saturation, whereas the free troposphere above can be extremely dry. Similarly, at the Upper Troposphere/lower Stratosphere boundary, the moisture is really low and this is critical for the ozone budget. However, downscaling the coarse vertical resolution is a different topic that could indeed be tackled with similar approaches, but requires different sets of proxies, and will be addressed in future work.

The paper is organized as follows. In section 2 we present the satellite data sources used in this study and in section 3 we discuss the physical background for our approach and its related limitations; section 4 describes the downscaling method used

to downscale water vapour observations from vertical cloud profiles; results are discussed in section 5 and finally, conclusions and future perspectives are drawn.

## 2 Data

### 2.1 SAPHIR

SAPHIR is a cross-track passive microwave sounder on board the Megha-Tropiques mission. It observes the Earth's atmosphere with an inclination of 20 degrees to the equator, a footprint size at nadir of 10x10 km$^2$, with a 1700-km swath made of scan lines containing 130 non-overlapping footprints (for more details see e.g. Brogniez et al. (2016) and references therein). SAPHIR provides indirect observations of the relative humidity ($RH$) in the tropics (28°S - 28°N) by measuring the upwelling radiation with six double-sideband channels close to the 183.3-GHz water vapour absorption. In this line of strong absorption

of radiation by water vapour, the measured radiation is affected both by the absorber amount (the water vapour) and the thermal structure, making the retrieval of $RH$ more straightforward and less dependent on a priori temperature or absolute humidity data (Brogniez et al., 2013).

   In this work, we used the layer-averaged $RH$ (six layers distributed between 100 and 950 hPa) derived by Brogniez et al. (2016), which is available for the period October 2011 - present. In this study, the authors adopted a purely statistical technique

to retrieve for each atmospheric layer the full distribution of $RH$ from the space-borne observations of the upwelling radiation and training $RH$ data derived from radiosondes profiles. This retrieval scheme was found to have similar performances compared to other methods that also rely on some other physical constraints (e.g. the surface emissivity, temperature profile, and a prior for $RH$ profiles for brightness temperature simulations). Figure 1a, shows an example, for each atmospheric layer, of the mean of the retrieved $RH$ distribution, derived as detailed in Brogniez et al. (2016).

Given the purpose of this study, we also note that the retrieval of $RH$ from the microwave sounder SAPHIR is not biased by the presence of ice particles, as soon as the ice crystals are small enough not to scatter the microwave radiation (Burns et al., 1997). Situations with large ice crystals, such as those produced during strong convective events, are discarded during the processing of the SAPHIR measurements (Brogniez et al., 2016).

### 2.2 CALIPSO

The lidar profiles in the GCM-Oriented Cloud-Aerosol Lidar and Infrared Pathfinder Satellite Observations (CALIPSO) Cloud Product (CALIPSO-GOCCP, Chepfer et al. (2010)), are designed to compare in a consistent way the cloudiness derived from satellite observations to that simulated by General Circulation Models (GCMs, (Chepfer et al., 2008)). CALIPSO-GOCCP product is available for the period June 2006 - December 2018. CALIPSO is a nearly sun-synchronous platform that crosses the equator at about 01:30 LST (Winker et al., 2009) and carries aboard the Cloud-Aerosol LIdar with Orthogonal Polarization

(CALIOP). CALIOP accumulates data of the Attenuated Backscattered (ATB) profile at 532 nm over 330 m along track with a beam of 90 m at the Earth's surface. The lidar scattering ratio ($SR$) is measured relative to the backscatter signal that a

molecular atmosphere (without clouds or aerosols) would have produced. Within a cloud the SR value represents a signature of the amount of condensed water within each layer convoluted with the optical properties of the cloud particles that depend on their size and shape. Values of $SR$ greater than five are taken as indications of layers containing clouds (Fig. 1b, see Chepfer et al. (2010) for more details). On the other hand, values of $SR$ lower than 0.01 correspond to layers that are not documented by CALIPSO. Indeed, layers located below clouds opaque to radiations are not sounded by the laser (Guzman et al., 2017; Vaillant de Guélis et al., 2017).

Following Chepfer et al. (2010), layers corresponding to values located below the surface ($SR = -888$), rejected values ($SR = -777$), missing values ($SR = -9999$) and noisy observations ($-776 < SR < 0$) were all set to missing. Moreover, in order to reduce the noise and the number of missing data, each $SR$ profile (40 equidistant layers with height interval of 480 m) was averaged as the following: in the boundary layer (below 2 km), the original vertical spacing was used (four layers in total), while, above, the layers were averaged every 1 km, giving in total $p = 21$ vertical layers. Only the averaged $SR$ profiles without any missing layer were retained: the choice of setting to missing all noisy layers, implies retaining mostly night-time data only (after excluding the averaged profiles with missing layers, the percentage of day-time profiles dropped from about 50% to less then 15%).

## 3   Physical approach and related limitations

Among the clouds forming in the troposphere, tropical ice clouds are of particular interest, because of their extensive horizontal and vertical coverage and their long lifetime (Sassen et al., 2008), and above all because they are intimately related to water vapour (Udelhofen and Hartmann, 1995).

This work is based on the following physical assumption: the small scale cirrus cloud properties (microphysics and contours) variations interplay with the small scale relative humidity (mixed of water vapour amount and temperature) variations. Indeed, cirrus clouds are composed of ice crystals, and ice crystal microphysical processes, such as nucleation, growth, evaporation depends on the presence of ice nuclei, water vapour amount and local cold temperatures. As a consequence, the latter influence the cloud contours, the density of the ice crystals within the cirrus clouds as well as the ice crystal sizes and shapes. These ice microphysical processes are embedded into large scale atmospheric circulation and into local dynamical motions.

In this study, we rely on the physical interplay between the small scale variations in the cloud properties (microphysics and structure/contours) and the small scale relative humidity variations, to downscale coarse observations of relative humidity to higher resolution (smaller scale).

For instance, at the microphysical scale the available water vapour is used for the growth of the ice crystals, which explains partly the drying of the upper troposphere during the formation of thin cirrus Jensen et al. (1996); Rosenfield (1998). Detrainment of moisture induced by the evaporation of droplets, yielding to situations of in-cloud supersaturation of water vapour, has also been highlighted around optically thick ice clouds (Krämer et al., 2017; Hoareau et al., 2016).

To characterize the small scale variation in the cloud properties (microphysics and cloud contours), we use clouds observations at high resolution (<500m) collected with the CALIPSO space lidar. CALIPSO does not directly observe the particle microphysical

properties, but it observes the lidar scattering ratio ($SR$) profiles that depends on the amount of condensed water, and therefore on a mix of concentration, size and shape of ice crystals in the atmosphere (Cesana and Chepfer, 2013) as stated in the standard lidar equation. $SR$ increases from 1 to 80 with the amount of condensed ice in the atmosphere, only when the cloud optical depth < 3 which is the case for most ice clouds. Indeed the variations observed in the values of the $SR$ are caused by small scale variations in the clouds properties: these variations are primarily driven by the ice crystal number concentration, and secondly by the variations in the phase (single phase or mixed phase), the shape and the size phase of the particles. In absence of clouds, the ice crystal number concentration is zero, and SR < 5, which delimits the contours of the cirrus cloud.

As there is an 'indirect correlation' between ice particle (shape, size, density, etc.) and $RH$, we can reasonably expect some correlation between $SR$ profiles from CALIPSO and water vapour profiles. For a given profile the vertical variation of $SR$ are modulated by the in-cloud variations in the vertical velocity, forced by large-scale dynamics, which affect the $RH$ through the condensation and the evaporation of cloud droplets (see Korolev and Mazin (2003), and references therein). Added together these properties influence and affect the surrounding $RH$.

Therefore, in the following, we assumed that the $RH$ retrieved from SAPHIR can be reasonably linked to ice clouds measured by CALIOP. Even further, we assumed that the measurements of ice clouds by CALIOP can be used to predict a particular $RH$ profile. Although the approach that we present in this study could in principle be extended to other cloud types, here we decided to focus on ice clouds over the ocean, for which the connection to water vapour is documented to be strong.

To avoid any misuse of the $RH$ high-resolution pseudo observations dataset built in this paper, we recall the reader that the small scale water vapour is not measured directly by the CALIOP lidar. The small scale water vapour is deduced from the indirect physical correlation between $RH$ and the lidar observations. For this reason, the high-resolution dataset of $RH$ pseudo-observations is not applicable for the following purposes: 1) to prove a correlation between water vapour and cloud observations from other lidar products 2) to prove a correlation between water vapour and cloud properties.

## 4   Methods

A three-step method was applied to downscale water vapour observations from vertical cloud profiles. First, we co-located SAPHIR and CALIPSO observations (section 4.1); then, using a statistical clustering technique, we selected only CALIPSO profiles corresponding to ice clouds 4.2), and finally we applied the downscaling method (section 4.3).

### 4.1   SAPHIR-CALIPSO co-location

To identify the times and locations where the orbits of SAPHIR and CALIPSO overlap, we first extracted all the observations at nadir falling within a distance of 50 km and within 30 min (for details of the software used for the co-location of the orbits see http://climserv.ipsl.polytechnique.fr/ixion). SAPHIR measurements (both at and off-Nadir) corresponding to the selected orbits were then matched to CALIPSO observations falling within each SAPHIR pixel, defined as the 10 km circle around its geographical coordinates (see Fig. 1c). In the following analysis, each SAPHIR measurement at coarse resolution

$(M = 1, ..., N)$ encapsulates $n(M)$ CALIPSO observations at fine scale $(m = 1, ..., n(M))$, where $n(M)$ changes depending on the spatial alignment of the two satellites. Figure 2 shows a sample of co-located CALIPSO and SAPHIR profiles. For SAPHIR measurements both the mean and the standard deviation of the retrieved distribution are shown. As Fig. 2c shows, larger uncertainties in the retrieved $RH$ are expected at lower altitudes because of the distribution of the sounding channels of
the radiometer and because of their bandwidth (Clain et al., 2015). The latter is narrow (0.2 GHz) for the central channels of the 183.31 GHz absorption line, which translates into a low uncertainty for the upper tropospheric estimates, and it stretches (2 GHz) for the channels located in the wings of the line, implying a larger uncertainty for the retrieval. In this study, we did not account for errors in the $RH$ retrieval (we used the mean of the $RH$ distribution from the retrieval algorithm) but this point can be further developed in future studies.

## 4.2   Selection of tropical ice cloud profiles

In order to select only profiles characterized by tropical ice clouds, the co-located samples were separated into clusters based on indicators of the type of clouds present at the moment of the observation.

The clusters were obtained by a $k$-means unsupervised classification of the reconstructed $SR$ profiles (e.g. Lloyd (1982)), rather than using the cloud phase flags associated with each vertical level as defined in Cesana and Chepfer (2013) (e.g. a
profile corresponding only to clear-sky and liquid observations is classified as LIQUID, see caption in Fig. 3 for more details). In fact, by averaging the $SR$ profiles above the boundary layer to a 1 km resolution with the aim of reducing the noise and the amount of missing data, we also had to apply the same averaging procedure to the cloud phase flag profiles in order to maintain a coherence between the $SR$ profiles used in the regression model and the corresponding cluster.

The reason of using a statistically-based clustering approach is twofold. First, the 'mixed' flags resulting from the averaging
procedure, requires some physical interpretation of these mixed pixels (e.g. are ICE-MIX, ICE-LIQ-MIX profiles representing the same vertical cloud structure?), while a statistically-based clustering method encompasses this problem. Additionally, by using the $k$-means approach, which allows to increase the number of clusters, the method might be better generalizable to boundary layer clouds. The latter are in fact characterized by a much larger variety in the $SR$ vertical structure (c.f. Fig. 2), which leads to more varied profiles (not shown) when using a global cloud flag that does not account for the order of the pixel
values.

Prior to clustering, and for clustering only, in order to further reduce the noise in the $SR$ profiles, these were transformed using a Principal Component Analysis (PCA, von Storch and Zwiers (1999)) analysis, where 90% of the variance was retained. Moreover, since layers with $SR$ values in the same range are associated to the same micro-physical properties, the reconstructed $SR$ profiles were first binned according to the interval boundaries suggested in Chepfer et al. (2010), as detailed in Fig. 5
in their study. Given an optimal number of clusters ($k$), this method partitions the observations into $k$ clusters with each observation belonging to the cluster with the nearest mean by minimizing the within-cluster-sum of squares ($wss$). Since the initial assignment of the observations to a cluster is random, the algorithm is run several times (here 100) and the partition with the smallest $wss$ is chosen amongst the different ensemble members. However, when $k$ is not known a priori, it must be selected from a range of plausible values (here: $k \in \{2, ..., 15\}$), and chosen so that adding another cluster does not produce a drastic

decrease in $wss$, and therefore does not improve significantly the quality of the clustering. For example for reconstructed $SR$ profiles in July 2013 over the Indian Ocean, this criterion yields between 8 and 13 clusters (not shown).

As Fig. 3 shows, both clusters named '1' derived by $k$-means with $k$=8 and $k$=13 show a similar mean $SR$ profile, with layers classified as cloudy mostly in the upper troposphere. As a further check that these profiles correspond indeed to ice clouds, we compared the $k$-means result with the clusters derived by combining the cloud phase flags associated with each vertical level. As Fig. 3 shows, again a similar characteristic $SR$ profile is observed for the cloud phase flag-based profiles classified as ICE/ICE-MIX.

This is further confirmed by the analysis of the distance between the mean $SR$ profile for each $k$-means-derived cluster and that classified by the ICE/ICE-MIX phase flag, which was found to be the smallest for the clusters named '1' for both $k$=8 and $k$=13. The distance was computed as the weighted Euclidean distance between each pixel of the mean $SR$ $k$-mean-derived profile and the corresponding pixel in the mean ICE/ICE-MIX $SR$ profile, with weights defined by the presence/absence of clouds (we used unitary weights if both pixels where cloudy ($SR$>5) and a weight of 9999 otherwise).

Therefore, in the following, the $k$-means classification is used to select all SAPHIR-CALIPSO co-located observations belonging to $SR$ clusters characterized by this typical mean $SR$ profile (in Fig. 3, clusters outlined by a red square).

### 4.3 Downscaling of water vapour measurements from cloud profiles

Given the SAPHIR-CALIPSO co-located samples belonging to ice cloud-type clusters as derived in the previous section, SAPHIR relative humidity at the $l$-th pressure level ($RH_l$, here corresponding to the mean of the distribution in Brogniez et al. (2016)) can be estimated in terms of an unknown function $\Phi$ of the $SR$ profile

$$RH_l \sim \Phi(SR_1, SR_2, ..., SR_p) \tag{1}$$

where $SR_1, SR_2, ..., SR_p$ designate $SR$ at each altitude level ($p$ = 21, following the vertical averaging implemented as described in section 4.2) and here represent the covariate data sources, also known as predictors. The method to downscale SAPHIR observations of relative humidity from CALIPSO $SR$ profiles consists in a two-stage regression model implemented directly on the observed spatial resolution (Liu and Pu, 2008; Malone et al., 2012). First, $RH_l$ is estimated based on the chosen statistical regression model (section 4.3.3). Secondly, the same regression model is applied iteratively to the predictions $\widehat{RH_l}$ and at each iteration step the multi-site results are corrected to harmonize the average of the estimates at fine resolution with its value at coarser scale (section 4.3.2).

#### 4.3.1 Choice of the regression model

The aim of this section is to compare different regression models for $RH_l$ given the set of predictors $SR_1, SR_2, ..., SR_p$ and to select the model with the 'best' predictions in a sense that will be clarified later. The models tested in this study are summarized in Table 1.

Random Forests (RF, Breiman (2001)), similarly to other machine learning techniques, does not require to specify the functional form of the relationship between the response variable and the predictors and, provided a large learning sample, has been shown to perform well (Hastie and Tibshirani, 2009) in the context of prediction of a response variable even with a non-linear relationship with a set of predictors. RF belongs to the family of classification and regression decision trees (Breiman

et al., 1984). Decision trees split the predictor space into boxes (or leaves) such that the homogeneity of the corresponding values of the response variable in each box is maximized. For regression trees, the homogeneity is defined as the sum of the residual-sum of squares ($rss$) with respect to the mean of the response variable within each box. As described in detail for example in Hastie and Tibshirani (2009), this method is implemented by sequentially splitting the predictor space into the regions $x_i < c$ and $x_i \geq c$ where the predictor $x_i$ and the cutting-point $c$ give the greatest possible reduction in $rss$. This binary

split is repeated until a minimum number of observations in each leaf is reached or because of an insufficient decrease in $rss$. Another possibility, which prevents overfitting, is to grow a tree with a large number of leaves but prune it at each split by controlling the trade-off of between the tree complexity (i.e. the number of leaves) and the fit to the data. Finally, the model estimate of the response variable is given by the mean of all the observations in each terminal leaf and for predictions for a new set of values of the predictors, one has then simply to follow the path in the tree until the final leaf is found. In order to reduce

the variance in the predictions, Breiman (1996) proposed to grow a tree on several bootstrapped samples of the original data and then take the average result from the different trees (*bagging*). This approach is justified by the property that by taking the average of $N$ independent observations with variance $\sigma^2$ we reduce the variance by $\sigma^2/N$. To avoid overfitting, the number of bootstrapped samples and that of the corresponding trees can be adjusted, while the trees are not pruned. With RF, the variance in the predictions can be even further reduced by retaining at each split a random selection from the full set of predictors,

therefore reducing the correlation between the trees generated by bootstrapping only.

*Bagging* and RF only estimate the conditional mean of the response variable but not its distribution, which can give information on the uncertainty in the predictions. On the other hand, Quantile Regression Forests (QRF, Meinshausen (2006)), by computing the Cumulative Distribution Function (CDF) of the response variable in each terminal leaf instead of its mean, represent a straightforward extension of the RF method, allowing to estimate any quantile of the response variable.

Non-parametric methods, like RF and QRF, do not allow to specify the functional form of the relationship between the response variable and the predictors. For this reason, we also tested the results obtained with a Generalized Additive Model (GAM, Hastie and Tibshirani (1986)), which is a statistical semi-parametric regression technique. A GAM is a Generalized Linear Model (GLM) with predictors involving a sum of non-linear smooth functions:

$$g\left(E\left[y|\mathbf{x}\right]\right) = \sum_{i=1}^{p} f_i\left(x_i\right) + \varepsilon \tag{2}$$

where $g(\cdot)$ is a link function between the expectation of the response variable $y$ (here the $RH$ of an atmospheric layer $l$) conditionally on a set of $p$ predictors $x_1, ..., x_p$ (here $SR_1, ..., SR_p$) and a sum of unknown univariate smooth functions of each predictor, $f_i(\cdot)$. $\varepsilon$ represents a zero-mean Gaussian noise. Here, $RH_l$ is assumed to follow a beta distribution, which is the usual choice for continuous proportion data, and its canonical link function, the logit $g(x) = \log\left(\frac{x}{1-x}\right)$, is used (Wood, 2011),

which assures that all values are in the (0,1) interval. To estimate each $f$, we can represent it as a weighted sum of known basis functions $z_k(\cdot)$

$$f(x) = \sum_k \beta_k z_k(x) \tag{3}$$

in such a way that Eq.(2) becomes a linear model, and only the $\beta_k$ are unknown. Here, we chose to represent the basis
functions as piecewise cubic polynomials joined together so that the whole spline is continuous up to second derivative. The borders at which the pieces join up are called knots, and their number and location control the model smoothness. To fit the model in Eq. (2), we used the approach of Wood (2011): the appropriate degree of smoothness of each spline is determined by setting a maximal set of evenly spaced knots (i.e. $bias(f) \ll var(f)$) and then controlling the fit by regularization, by adding a 'wiggliness' penalty $\int f''(x)dx = \boldsymbol{\beta}^T S \boldsymbol{\beta}$ to the likelihood estimation:

$$\mathcal{L}(\boldsymbol{\beta}) - \boldsymbol{\beta}^T \mathbf{S} \boldsymbol{\beta} \tag{4}$$

where $\mathcal{L}$ is the likelihood function of the $\boldsymbol{\beta}$ parameters and $\mathbf{S}$ the penalty matrix, with elements for the $k$th - $\widetilde{k}$th terms $S_{k\widetilde{k}} = \int z_k''(x) z_{\widetilde{k}}''(x)\, dx$.

Ideally, we would like to account for a neighbouring structure, i. e. neighbouring $SR$ profiles should be characterized by similar model parameters. This effect can be accounted for by assuming, under the Markovian property, that the model
parameters for the $m$th profile are independent of all the other parameters given the set of its neighbours $\mathcal{N}(m)$. This neighbouring structure can then be modelled by adding to Eq. (2) a smooth term with penalty

$$\Gamma(\boldsymbol{\gamma}) = \sum_{m=1}^{n} \sum_{\widetilde{m} \in \overline{\mathcal{N}(m)}} (\gamma_m - \gamma_{\widetilde{m}})^2 \tag{5}$$

where $\gamma_m$ is the smooth coefficient for region $m$ and $\overline{\mathcal{N}(m)}$ denotes the elements of $\mathcal{N}(m)$ for which $\widetilde{m} > m$. The penalty in Eq. (5) can be then rewritten as $\Gamma(\boldsymbol{\gamma}) = \boldsymbol{\gamma}^T \mathbf{S} \boldsymbol{\gamma}$ with $S_{m\widetilde{m}} = -1$ if $\widetilde{m} \in \mathcal{N}(m)$ and $S_{m\widetilde{m}} = n(m)$ where $n(m)$ is the number of
profiles neighbouring profile $m$ (not including $m$ itself). This specification is very computationally efficient, given the sparsity of the parameters precision matrix, and is known as Gaussian Markov random field (GMRF, Rue and Held (2005)). Here, we implemented this augmented model by defining two CALIPSO $SR$ profiles as neighbours if they belong to the same SAPHIR pixel.

Another possibility, although more computationally expensive, is to explicitly include in our model the spatial correlation
structure of the predictors by a fusion of geostatistical and additive models, known as geoadditive models (Kammann and Wand, 2003). These models allow accounting not only for the non-linear effects of the predictors (under the assumption of additivity) but also for their spatial distribution: two $SR$ profiles, and therefore the corresponding water vapour structures, are more likely to be dependent if they are close, by some metric. Given a set of geographical locations $\mathbf{s}$, a (bivariate) smooth

term $f(\mathbf{s})$ can be represented as the random effect $f(\mathbf{s}) = (1, \mathbf{s}^T)\boldsymbol{\gamma} + \sum_j w_j C(\mathbf{s}, \mathbf{s}_j)$ with $w \sim N(0, (\lambda C)^{-1})$, $\boldsymbol{\gamma}$ a vector of parameters and $C(\mathbf{s}, \mathbf{s}_j) = c(||x - x_j||)$ a non-negative function such that $c(0) = 1$ and $\lim_{d \to \infty} c(d) = 0$, which is interpretable as the correlation function of the smooth $f$ (Wood, 2011). By adding this term to the model in Eq. (2), we explicitly include the spatial autocorrelation in the $SR$ data without changing the mathematical structure of the minimization problem, and we can still use the GAM basis-penalty representation (Wood, 2011). Here, we assumed an isotropic exponential correlation function $C(\mathbf{s}, \mathbf{s}_j) = \exp(- \parallel \mathbf{s} - \mathbf{s}_j \parallel /r)$ with the range $r$ chosen equal to the size of SAPHIR pixels (10 km).

Following Ferro (2008), Ferro et al. (2014), and Taillardat et al. (2016), to assess the prediction skills of such models, scoring rules can be used to assign numerical scores to probabilistic forecasts and measure their predictive performance. Given an observation $y$, for a model ensemble forecast with members $x_1, .., x_K$ a fair estimator (Ferro et al., 2014) of the continuous ranked probability score (CRPS) is

$$CRPS(y) = \frac{1}{K} \sum_{i=1}^{K} | x_i - y | - \frac{1}{2K(K-1)} \sum_{i=1}^{K} \sum_{j=1}^{K} | x_i - x_j | \tag{6}$$

where lower values of the CRPS indicate better predictive skills. For regression techniques that estimate the conditional mean only (RF, GAM, GAM with GRMF, and the geoadditive method), the CRPS score accounts only for the accuracy of the forecast (the second term in Eq. (6) is zero), while for probabilistic methods, like the QRF method, it also accounts for the forecast precision. Typically, in order to directly compare a prediction system to a reference forecast (e.g. a climatology), the continuous ranked probability skill score (CRPSS) is needed

$$CRPSS = 1 - \frac{CRPS_{mod}}{CRPS_{ref}} \tag{7}$$

The CRPSS is positive if and only if the model forecast is better than the reference forecast for the CRPS scoring rule.

### 4.3.2 Iterative downscaling

Following the approach of Liu and Pu (2008) and Malone et al. (2012), the predictions were further optimized by ensuring that, for all layers, the observed relative humidity is as close as possible to the average of the predicted $RH$ distributions within the corresponding encapsulating SAPHIR pixel. This approach is meant to preserve the so-called 'mass balance' with the coarse scale SAPHIR information, and can be easily implemented with the following iterative approach:

1. within each SAPHIR pixel $(M)$, update the predictions $\widehat{RH}_l$: $\widetilde{RH}_l(m) = \widehat{RH}_l(m) + RH_l(M) - \frac{1}{n(M)} \sum_{j \in n(M)} \widehat{RH}_l(j)$

2. with the chosen regression model, regress the updated predictions $\widetilde{RH}_l$ with respect to the set of predictors $SR_1, SR_2, ..., SR_p$

3. if the coefficient of determination $(R^2)$ with respect to the observed relative humidity $RH_l(M)$ of the updated predictions is larger than that of the previous iteration than repeat steps [1]-[2], otherwise stop at previous iteration.

For ensemble models, like QRF, the update predictions and $R^2$ are computed on the median of the distribution only.

### 4.3.3 Remarks on the definition of the term downscaling

The downscaling scheme presented in this study differs from the classical downscaling approach where local variables, generally point-scale observations, are generated from large-scale variables, available at the much coarser grid-scale resolution typical of climate models and reanalyses outputs, and some point-scale covariate(s) at the same fine-scale spatial resolution as the response variable (e.g. elevation data). For this purpose, amongst other methods, also regression-based methods have been used (e.g., Vrac et al. (2007), where the model is trained on the available local variables, representing the ground truth. In this case, the evaluation of the fidelity of the downscaling is straightforward, as one can compare the predictions from the model to local observations that were not used for training (e.g., Vaittinada Ayar et al. (2015)).

However, in the case presented in this study, no $RH$ observations at the horizontal resolution of the cloud measurements (or higher) are available such that, when co-located with CALIPSO data, provide a large enough training or even testing set for the regression model. This means that in order to obtain some estimates of $RH$ that vary with cloud profiles, we are forced to the opposite approach, where the coarse $RH$ observations measured by SAPHIR are taken as the ground truth, and are regressed against the cloud profiles. Given that the cloud profiles are measured at finer resolution, we refer at the predictions derived in this way as downscaling, since we can incorporate the higher-resolution variability of the covariates in the estimates of the response variable.

In this context, without some additional independent validation with high resolution measurements, the accuracy of the predictions cannot be directly assessed since the model error cannot be quantified at the level of the finer-resolution observations. On the other hand, by adopting the QRF model, we are able to provide uncertainty estimates in the model predictions that account for the $RH$ variability (at the resolution of the coarse scale measurements), while applying the 'mass-balance' correction ensures the best possible consistency with the original measured values.

Clearly no point-to-point validation can be reasonably performed considering the time scales of *in-situ* or ground-based measurements vs. satellite measurements. However, it might still be possible to gain insights on the quality of the downscaling by statistically comparing the $RH$ distributions from available higher-resolution instruments (e.g. water vapour profiles from lidar collected by recent airborne field campaigns) and the downscaled profiles derived with the method presented in this study. Nevertheless, this will require extending the method on all years and locations of available data as well as to other cloud types, which is beyond the scope of the present study.

The fact that within the framework presented in this study, at the finer resolution scale, the model error cannot be directly separated from the variability in the response variable, might create some confusion on the meaning of term downscaling as adopted here. Nonetheless, for the model estimates, the variance explained by the cloud profiles is, by construction, higher than that for SAPHIR measurements, and this serves as a justification for the downscaling term: the predictions from the model are better correlated with the higher-resolution cloud profiles, and can therefore be considered as a downscaled product, in the sense discussed above.

## 5 Results and discussion

Figure 4 shows, for ice cloud profiles in the Indian Ocean in July 2013 ($k$=8), the comparison of the CRPSS computed for the forecast derived for the different regression methods (QRF, RF, GAM, GAM with GRMF, and the geoadditive method) with respect to the reference CRPS computed from the empirical distribution of the observations. In order to validate the regression results with independent test data, the predictions were performed using a 5-fold cross validation scheme. However, in order to reduce the computation time, cross validation was limited to the first iteration step, as, at this point, we were interested in comparing the performance of the different models rather than performing the full downscaling. For the RF and QRF method, the sensitivity of the results to the model parameters (number of trees and number of predictors selected at random at each split) was also investigated using a grid search; however, for both models, variations in the prediction skills (both in terms of $R^2$ and the CRPSS score) were found negligible with respect to the choice of these parameters, that were therefore set to their default values (c.f. the `randomForest` R package, R Core Team (2017)). The largest CRPSS is obtained using the QRF method, with a median value larger than 0.5 for all layers. The $RH$ predicted with the RF method are also significantly better that what we would obtain from the empirical distribution of the observations, although the probabilistic approach taken in QRF is more skill-full. On the other hand, all GAM-derived methods have a lower score, with CRPSS median values overall below 0.5, although, apart from the highest and lowest layers, all medians are above zero. As the CRPSS reveals, full non-parametric methods that do not rely on any assumption on the probability distribution of the response and that are free to learn any functional form from the training data, perform significantly better.

A positive value of the CRPSS for all $RH$ layers indicates a high level of correlation along the full vertical profile, which is expected for ice clouds: within and in the neighbourhood of regions of deep convection, which is their primary source (Hartmann et al., 2001), air masses are rapidly transported from the boundary layer through the free troposphere into the tropopause region (Corti et al., 2006). This is also shown in Fig. 5, which shows the median of the distribution of the predicted $RH$ for each vertical layer using the QRF method vs. the $RH$ observed by SAPHIR (at 10x10 km resolution). Here the predictions are the results of the 5-fold cross validation procedure, and are therefore derived from a model trained on an independent part of the data set. For layers L1-L5, the data are distributed close to the identity line, with the model explaining a large proportion of the variance of the observed $RH$ ($R^2 \geq 0.7$). On the other hand, as expected for ice clouds which populate the upper troposphere, lower correlation values are found for the lowest layer (L6, $R^2 \sim 0.4$). It should be noted that although a comparison with other sources of $RH$ data could be interesting, it will not necessarily be a validation of the results of our model. In fact, a part from the difficulty of finding a statistically significant sample of, for example, radiosondes or airplane observations co-located in space and time with CALIPSO measurements, these sources are characterized by different spatial resolutions from lidar data, which makes the comparison not straightforward.

To assess the importance of the cloud structure on the predicted relative humidity at different layers, we can compute, for each predictor, the decrease in accuracy obtained by randomly permuting its values (Fig. 6): the larger this value, the more important a predictor is. For the higher layers, as expected, this metric highlights the larger contribution of $SR$ layers corresponding to layers classified as cloudy, which are observed above $\sim 10$ km (c.f. Fig. 3). On the other hand, for layers closer to the surface,

the contribution of lower, (on average) non-cloudy $SR$ layers is found to be equally important because of the moisture that originates over warm waters.

Finally, as Fig 7 shows, the CRPSS distribution is similar for different choices of clusters (k-means with $k=8$ and $k=13$ and for the cluster corresponding to profiles with ice cloud pixels only) as well as for different seasons (July and January) and regions (Indian Ocean and Pacific Ocean): for all the layers the median CRPSS is positive, which confirms the robustness of the approach. These results are also independent (not shown) on the temporal difference and the spatial alignment of the co-located samples, on the distance from the coast, or on the uncertainty (standard deviation) in the observed relative humidity by SAPHIR.

Overall, these results suggest that, at the instantaneous scale of cloud measurements, the water vapour response along the whole troposphere in correspondence with ice cloud profiles is well predicted only accounting for their capability to backscatter radiation (given by the observed $SR$ profile). While the large-scale link between relative humidity and the cloud properties (vertical distribution, phase and opacity) has been well documented in previous studies (Martins et al., 2011; Reverdy et al., 2012), this work represents the evidence that this relationship can also be detected at much smaller spatio-temporal scales. The emergence of a clear signal at these fine scales, also highlights the limitations of SAPHIR measurements: although SAPHIR observes the water vapour field at a much finer horizontal resolution than what is currently available in reanalysis products, in order to explain physical processes, downscaled observations are needed. Figure 8 compares, for a selection of ice cloud profiles ($n(M) > 25$), the corresponding layers of relative humidity observed by SAPHIR with the median of the downscaled results derived by implementing the iterative QRF scheme. For all layers, the iteration typically stops after 2-3 steps and, although increases the $R^2$ between SAPHIR observations and the predicted relative humidity by only few percent, ensures consistency with the observed data, as described in section 4.3.2. The goal of the downscaling scheme implemented in this work is to reconstruct the variation of the relative humidity field at the fine resolution of cloud measurements within each SAPHIR coarsely resolved pixel: as Fig. 8 shows, the downscaled values exhibit variations within the same SAPHIR pixel depending on the corresponding $SR$ profile (Fig. 8c) that cannot be observed by SAPHIR (Fig. 8b). As discussed at the beginning of this section, a measure of the reliability of these variations can be derived from the spread of the predicted distribution, given here as the interquartile range (Fig. 8d). Differences between the downscaled and the observed $RH$ observations will be larger when the $RH$ field is characterized by finer-scale heterogeneities deriving from finer-scale processes, as for instance Fig. 8e seems to suggest for some of the profiles. However, these differences are expected since with the method presented here the predicted relative humidity structure incorporates the higher-resolution variability from cloud profiles. On the other hand, as shown both in Fig. 4, 5, and 7, the downscaling model is able to successfully explain the coarse-scale $RH$ observations from the finer-scale $SR$ measurements and the overall bias is low, which gives us confidence in the predictions.

The intra-pixel $RH$ variations are further analysed in Fig. 9, which shows for a single SAPHIR pixel overlaid on the observed values, the downscaled predictions from the QRF and the geoadditive model. For the latter, the predictions were extended outside the observed CALIPSO locations on the direction orthogonal to CALIPSO track line up to 1 km on each side. The relative humidity field at these new locations was predicted using the model fitted through the iterative scheme for the available CALIPSO observations and assuming that each $SR$ profile was also representative of the cloud distribution for

locations shifted along the direction orthogonal to CALIPSO track within a distance of 1 km. As expected and shown by Fig. 9b, the largest part of the variance is explained by the $SR$ predictors, while variations related to the spatial smooth are almost not noticeable with the scale used in the plot, compared to the variations in the predictions for a given $SR$ profile. In other words, once the effect of the SR predictors is taken into account, the residuals (i.e. the difference between the observed

and the predicted RH) do not show spatial autocorrelation. This has the counter-intuitive effect that each pixel also seems representative of the pixels in the direction orthogonal to the flight direction (where cloud observations are not available) while showing strong variations in the flight direction. However, this does not imply that there are no variations to the side of each pixel. Instead, what this result shows, is that the model is not improved by accounting for any residual spatial random effect.

Although the CRPSS quantifies the quality of the predictions (w.r.t. the climatology) conditionally on the regression model

and the predictors, for direct validation, observations of relative humidity at the scale of the clouds measurements would be required. In principle, the network of radiosonde measurements, which provides $RH$ quality-checked data (Durre et al., 2006) and has been used in previous studies for validation of satellite measurements, including SAPHIR (Sivira et al., 2015; Brogniez et al., 2016), could be used for validation purposes. However, in practice, its limited spatial coverage, with also most of the observations falling over land, hampers the feasibility of this approach. On the other hand, probabilistic approaches, like the

QRF method, by assessing the uncertainty in the predictions through the spread of the distribution, allow the quantification of the confidence in those predictions and therefore, in a way, provide an indirect estimate of their quality.

## 6    Conclusions

We have presented a method to downscale observations of relative humidity ($RH$) available from the passive microwave sounder SAPHIR at a nominal horizontal resolution of 10 km to the finer resolution of 90 m using scattering ratio ($SR$)

profiles from the lidar CALIPSO. The method was applied to ice clouds profiles over the tropical oceans, where the connection to water vapour is expected to be stronger.

By using an iterative regression model of the satellite-derived $RH$ with the $SR$ profiles as covariates, we were able to successfully predict the relative humidity along the whole troposphere at the resolution of cloud measurements. The method also ensures that the average of the predicted $RH$ distributions within the corresponding encapsulating SAPHIR pixel is as

close as possible to the observed value. Amongst the different regression models tested, the best results were obtained using a Quantile Random Forest (QRF) method, with a coefficient of determination ($R^2$) with respect to the observed relative humidity larger than $0.7$ and a CRPSS with respect to the climatology with a median value larger than $0.5$ for all layers down to 800 hPa. High explanatory power along the full vertical profile is expected for ice clouds, for which deep convection, by transporting air masses from the boundary layer up to the tropopause region, is their primary source.

By providing a method to generate profiles of water vapour (at high spatial resolution) from simultaneous co-located cloud profiles, this work will be of great help to revisit some of the current key barriers in atmospheric science. While SAPHIR record only stretches back to 2011, CALIPSO cloud measurements are available since 2006, a period that includes three El Niño/Southern Oscillation (ENSO) cycles. A 10-year long high-resolution water vapour-clouds combined dataset might allow:

- to study how small scale water cycle processes behave when exposed to strong variations in large scale circulation regimes such as those associated to El Niño cycles

- to 'evaluate' how small scale water vapour inhomogeneities affect the water vapour simulated by standard reanalyses (e.g. ERA-Interim Dee et al. (2011), NCEP Kalnay et al. (1996), etc.), which are known to badly parameterize clouds and to have biases in water vapour in the upper troposphere (Jiang et al., 2015; Davis et al., 2017; Schröder et al., 2017)

- to put the results of past and current field experiments into a larger scale context, e.g. identifying if results of specific campaigns are representative of large portions of the tropical belt

- to guide the parametrization of unresolved subgrid-scale water vapour/clouds processes to reduce cloud feedback uncertainties (Randall et al., 2003) in climate models which ultimately will contribute to improve model-based estimates of climate sensitivity

- to evaluate the description of water vapour/cloud interactions in regional models - e. g. WRF, Meso-NH (Chaboureau et al., 2002; Fan et al., 2007), which although having a fine-enough grid-spacing to allow explicit simulations of the mesoscale dynamics associated with convective clouds (Guichard and Couvreux, 2017) still integrate parametrizations to represent sub-grid-scale motions, micro-physics, and radiative processes

- to test the validity of the fixed anvil temperature hypothesis (Hartmann and Larson, 2002) and estimate the changes to long-wave fluxes with warming, for example using simulated CALIPSO profiles from model variables (Chepfer et al., 2008)

- to quantify the limits of current and future space missions by characterizing the spatial inhomogeneities in water vapour fields that cannot be observed by present satellites and will likely not be observed within the next two decades (e.g. 2017-2027 "Decadal Survey for Earth Science and Applications from Space") due to technological limits.

We also note that the method developed in this study will be extended to other types of clouds, although additional covariates might be required. In fact, while SAPHIR is not able to retrieve the $RH$ profile in the case of heavy precipitation, which implies that the majority of ice clouds co-located with SAPHIR measurements are non-precipitating, this is not true for light precipitating clouds, which typically correspond to low-level liquid clouds only. Therefore, for liquid clouds, including the radar reflectivity as measured by the radar CloudSat, which is indicative of the intensity of rainfall, might increase the model explanatory power.

Finally, the downscaling method presented here could be also applied to other satellite products, with the underlying assumption of using covariate data that are strongly related to the target variable. For example, this same method using CALIPSO $SR$ profiles as predictors, can be applied to downscale the precipitation observed by CloudSat, for which small scale observations at global scales are not available.

*Data availability.* A sample dataset of simultaneous co-located scattering ratio profiles of tropical ice clouds and observations of relative humidity downscaled at the resolution of cloud measurements is publicly available and can be freely downloaded at http://dx.doi.org/10.14768/20181022001.1 (Carella et al., 2019).

*Author contributions.* GC developed the methodology, and drafted the manuscript. MV, HB, PY, and HC supervised and supported the development of the methodology and provided detailed comments on the manuscript.

*Competing interests.* The authors declare no competing interest.

*Acknowledgements.* The authors are thankful to Patrick Raberanto (Laboratoire de Météorologie Dynamique) for his help with the co-location of SAPHIR and CALIPSO orbits. The authors would like also to thank the IPSL mesocenter and ESPRI teams from IPSL for providing computing and storage resources, and CNES and NASA for providing SAPHIR and CALIPSO Level 1 data. GC was supported by the Paris-Saclay Initiative de Recherche Strategique SPACEOBS (ANR-11-IDEX-0003-02). The authors also acknowledge the support of CNES, program EECLAT.

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

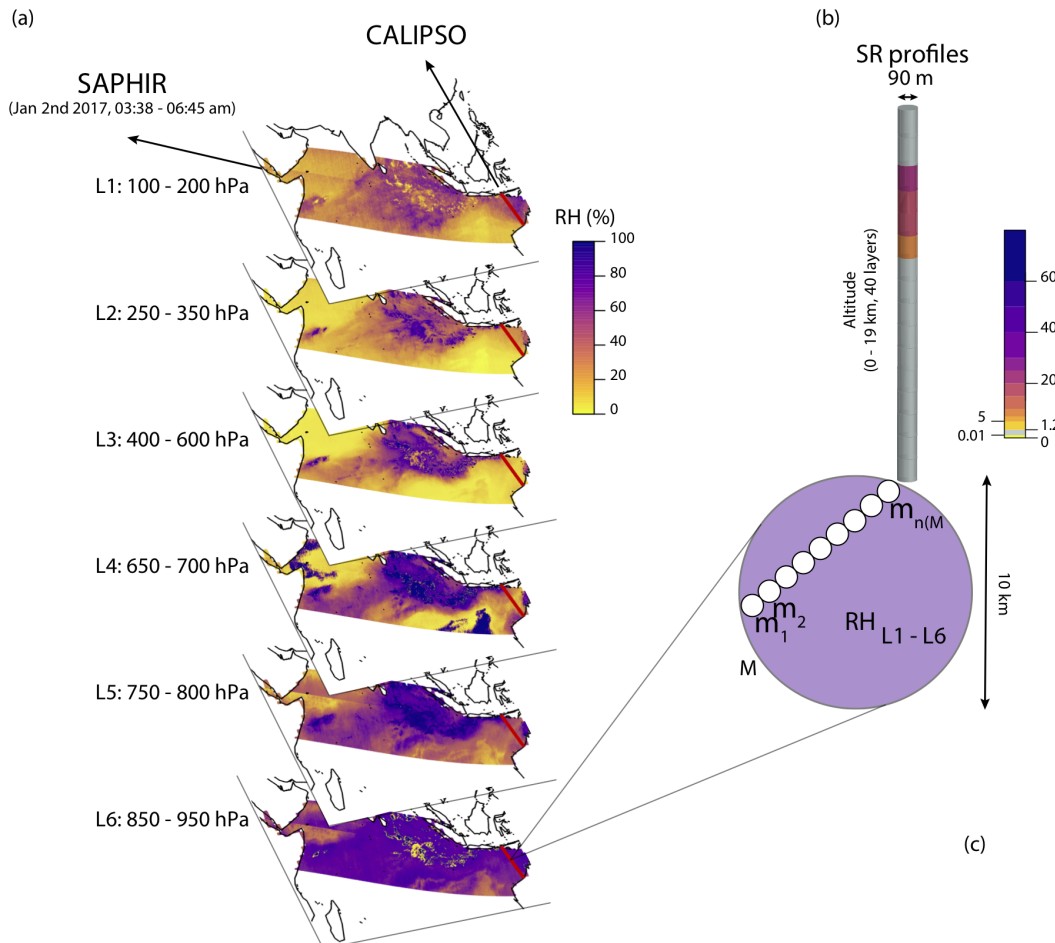

**Figure 1.** (a): $RH$ (mean) observed by SAPHIR for all six pressure layers, in the Indian Ocean on January 2nd 2017 between 03:38 and 06:45 am. Overlaid is the CALIPSO track line (*red line*). (b): example of $SR$ profile measured by CALIPSO. (c): schematic representation of SAPHIR-CALIPSO co-location: $M = 1, ..., N$ SAPHIR measurements at coarse resolution encapsulating $m = 1, ..., n(M)$ finely-resolved CALIPSO observations.

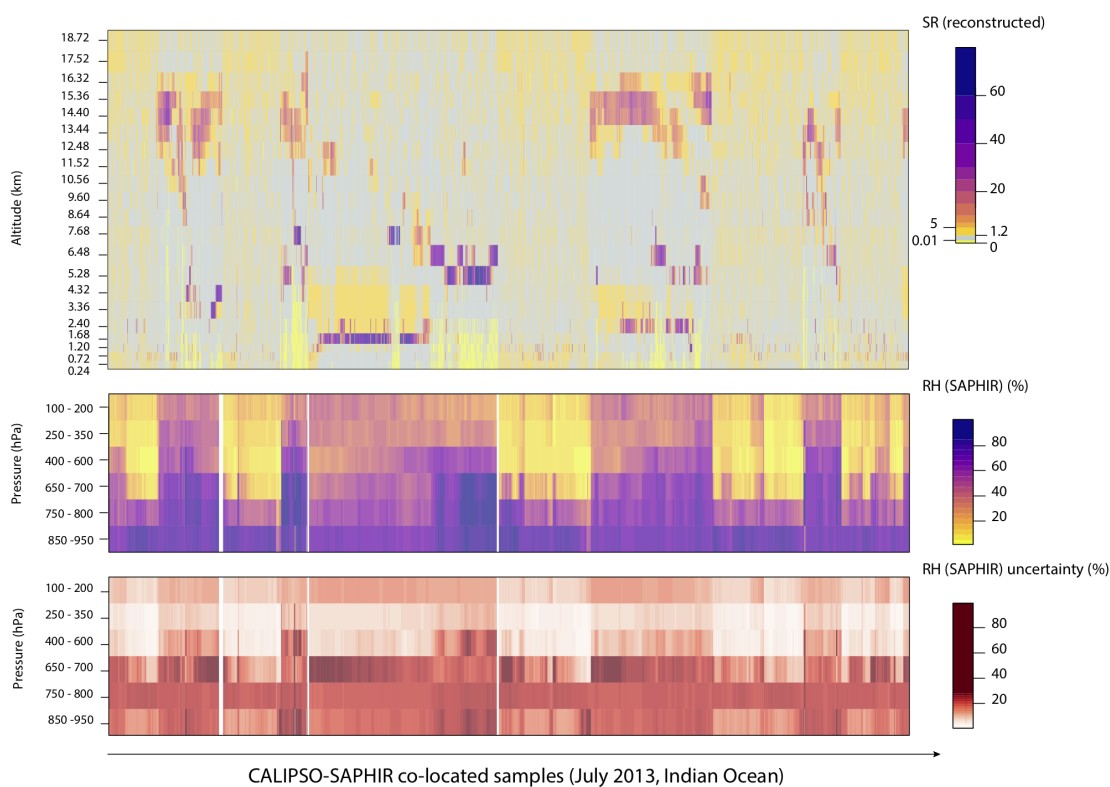

**Figure 2.** Reconstructed $SR$ profiles for a selection of CALIPSO samples in the Indian Ocean, July 2013 (*top*) and co-located $RH$ observations from SAPHIR (mean and uncertainty (standard deviation), *middle and bottom*). As in Chepfer et al. (2010), $SR > 5$ correspond to cloudy observations, $0 < SR < 0.01$ (*light yellow*) correspond to fully attenuated observations, $0.01 < SR < 1.2$ (*grey*) correspond to clear sky, and $1.2 < SR < 5$ (*dark yellow*) correspond to unclassified observations. Note that the reconstructed $SR$ were only used for layers indicating clouds to avoid mixing of cloud and clear sky values. The x-axis represents the co-location index. Overall, $RH$ measurements with a standard deviation larger than 30% might be considered very uncertain.

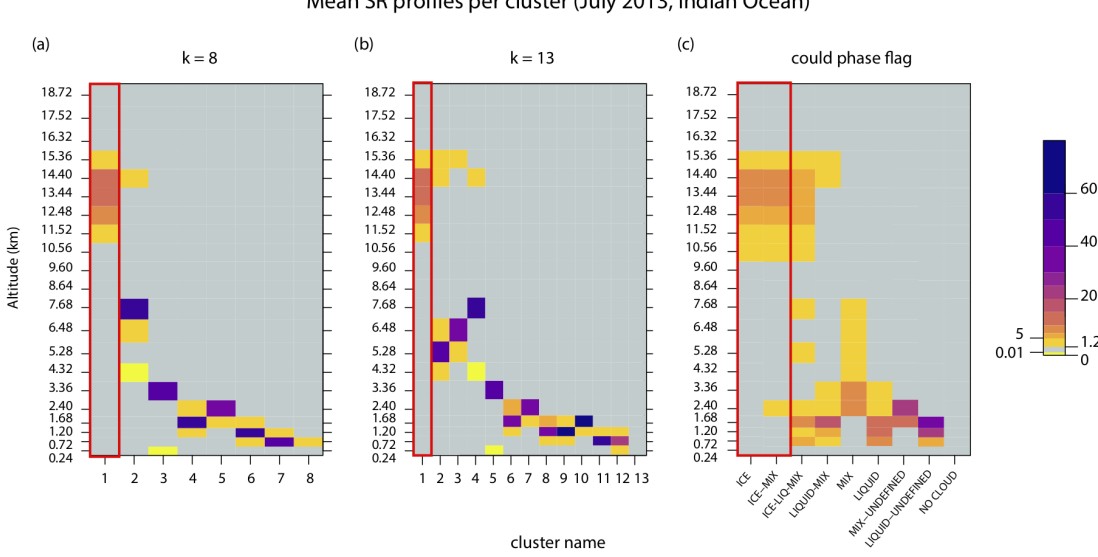

**Figure 3.** Mean $SR$ profile per cluster for different choices of the clustering method (Indian Ocean, July 2013). (a): Mean $SR$ profile per cluster obtained by a $k$-means classification setting $k$=8. (b): as (a) but setting $k$=13. (c): Mean $SR$ profiles per cluster derived by combining the cloud phase flags in Cesana and Chepfer (2013). ICE: observations classified as ice only. LIQUID: observations classified as liquid only. MIX: profiles containing $SR$ values derived by averaging observations classified as liquid and observations classified as ice. UNDEFINED: observations for which the cloud phase flag in Chepfer et al. (2010) is 'undefined', 'horizontally oriented' or 'unphysical'. The cluster type is then defined as the combination of these flags. Profiles characterized by other combinations of flags (e.g. FALSE LIQUID, FALSE ICE, etc.) correspond to less than 250 observations and have been omitted. Selected anvil-type clusters are outlined by a *red square*.

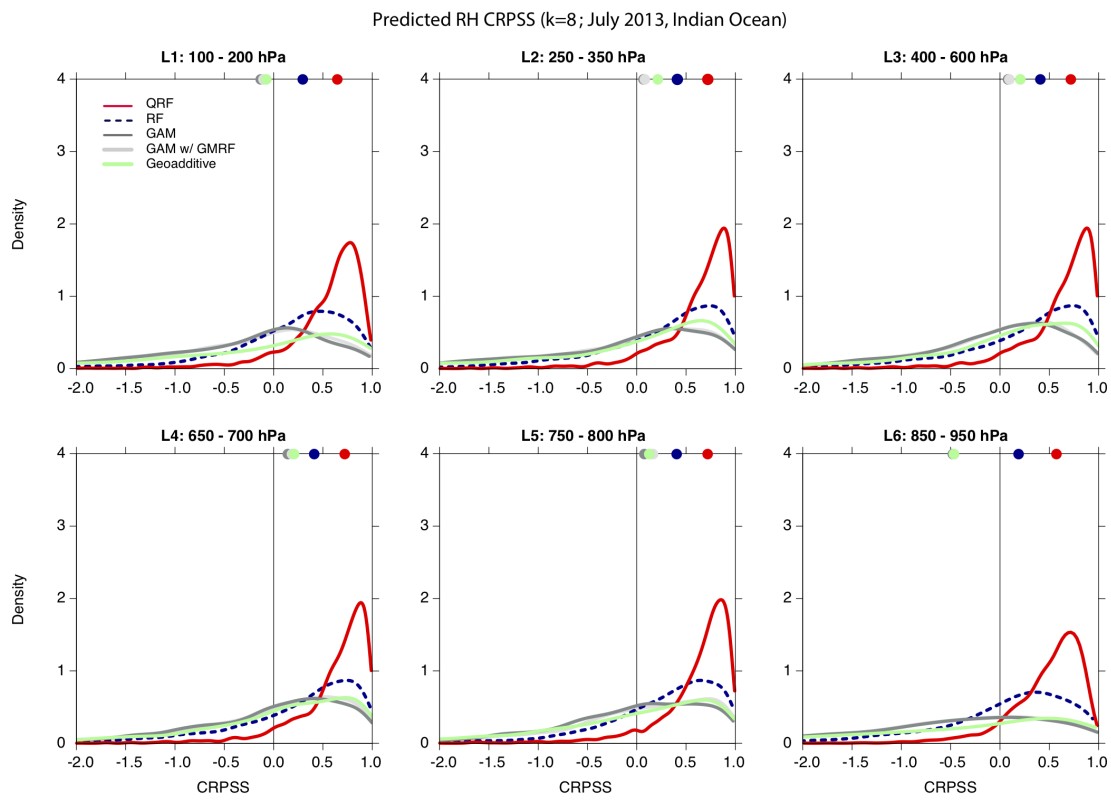

**Figure 4.** CRPSS score for ice cloud profiles (*k*=8) in the Indian Ocean, July 2013: QRF (*red solid line*), RF (*blue dashed line*), GAM (*dark grey solid line*), GAM with GMRF smoother (*light grey solid line*) and with the geoadditive method (*green solid line*). The dots at the top of each panel indicate the median of the distribution. Predictions are from the validation set within a 5-fold cross validation scheme.

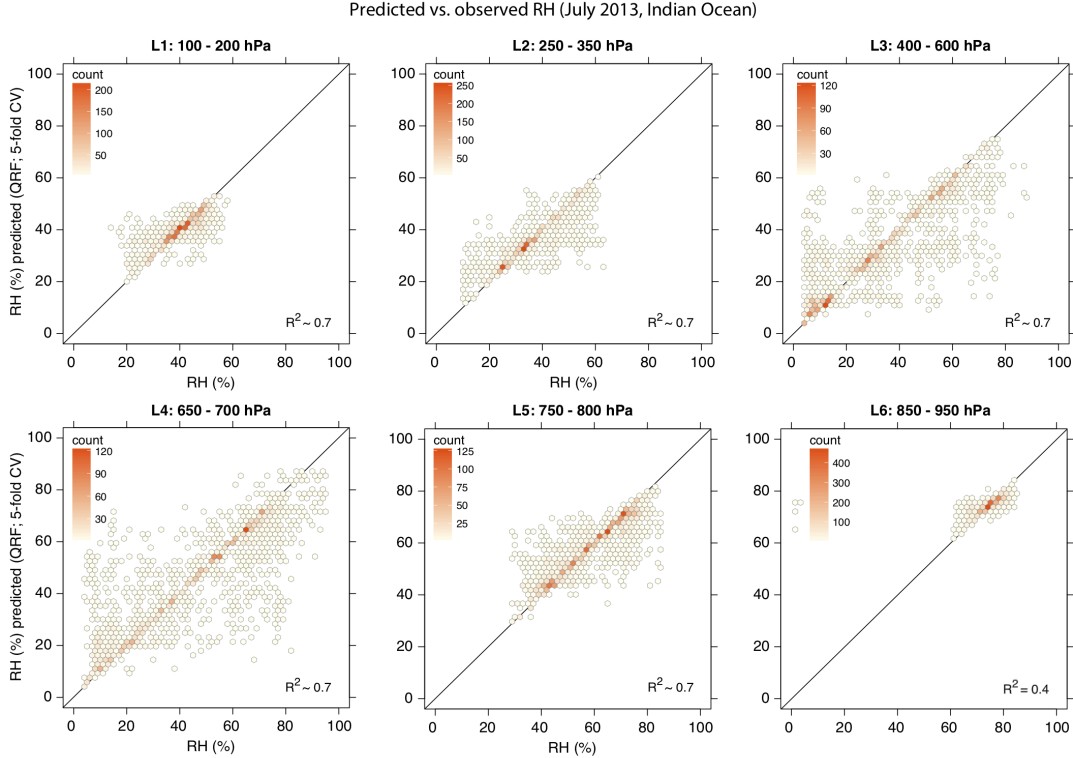

**Figure 5.** Scatter plot of the median of the predicted distribution vs. observed $RH$ for ice cloud profiles ($k$=8) in the Indian Ocean, July 2013. Predictions are made using the QRF method and are from the validation set within a 5-fold cross validation scheme. $R^2$ is computed as $1 - \dfrac{\sum_i (y_i - \widehat{y_i})}{\sum_i (y_i - \overline{y})^2}$ where the $y_i$ represent SAPHIR observations with mean $\overline{y}$ and $\widehat{y_i}$ are the cross-validation predictions.

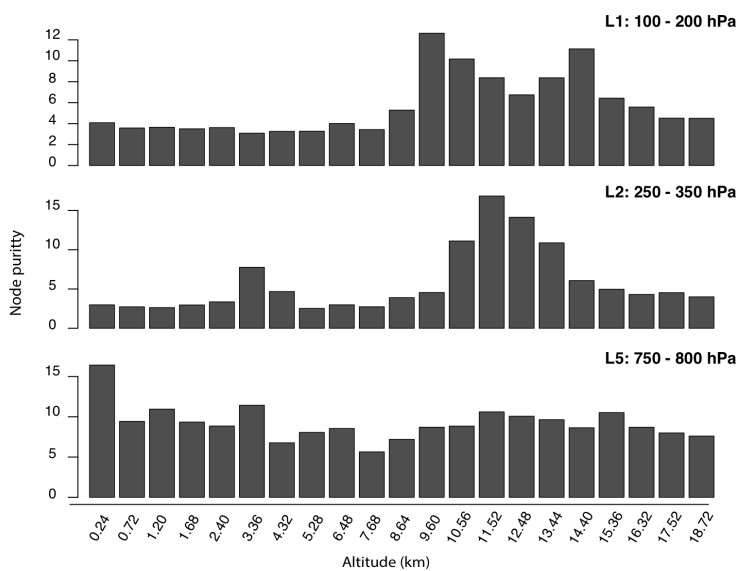

**Figure 6.** Variable importance (QRF method) for the predicted $RH$ for ice cloud profiles ($k$=8) in the Indian Ocean, July 2013.

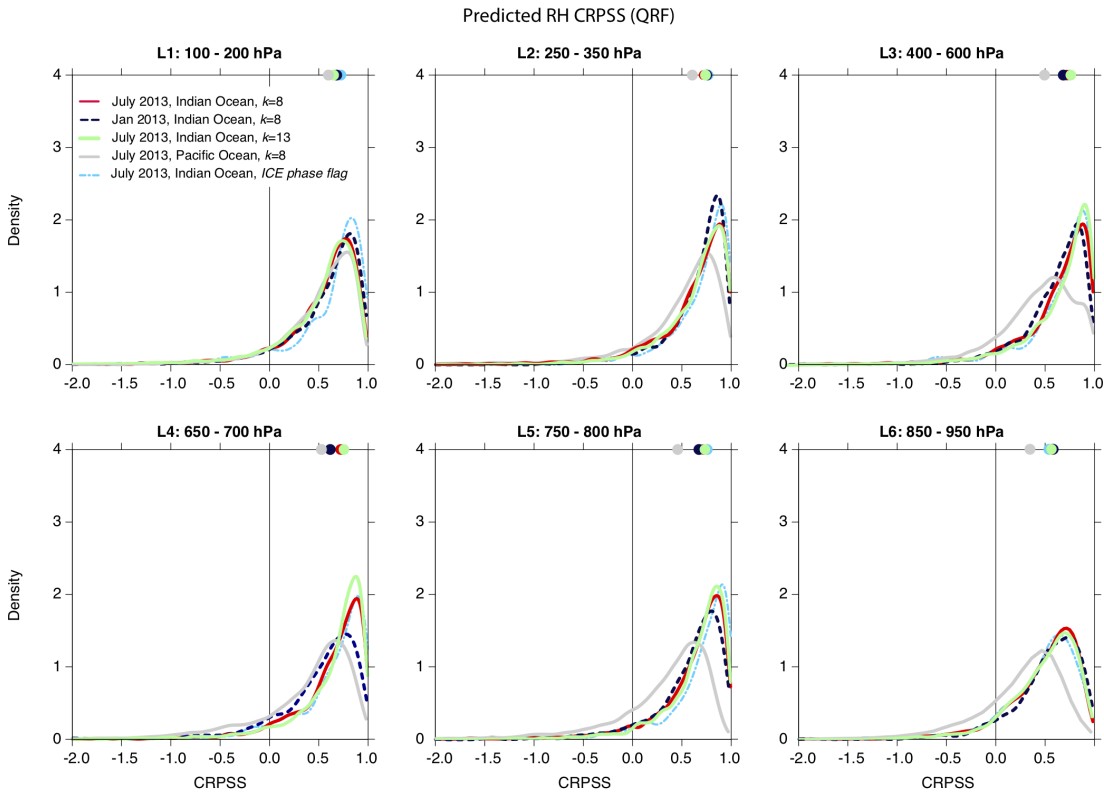

**Figure 7.** CRPSS score for ice cloud profiles (QRF method): Indian Ocean, July 2013 for k-means derived clusters setting *k*=8 (*red solid line*), *k*=13 (*dark blue dashed line*), and for cloud phase flag-based profiles classified as ICE (*light blue dot-dashed line*); Indian Ocean, January 2013 setting *k*=8 (*dark grey solid line*); Pacific Ocean, July 2013 setting *k*=8 (*light grey solid line*). The dots at the top of each panel indicate the median of the distribution.

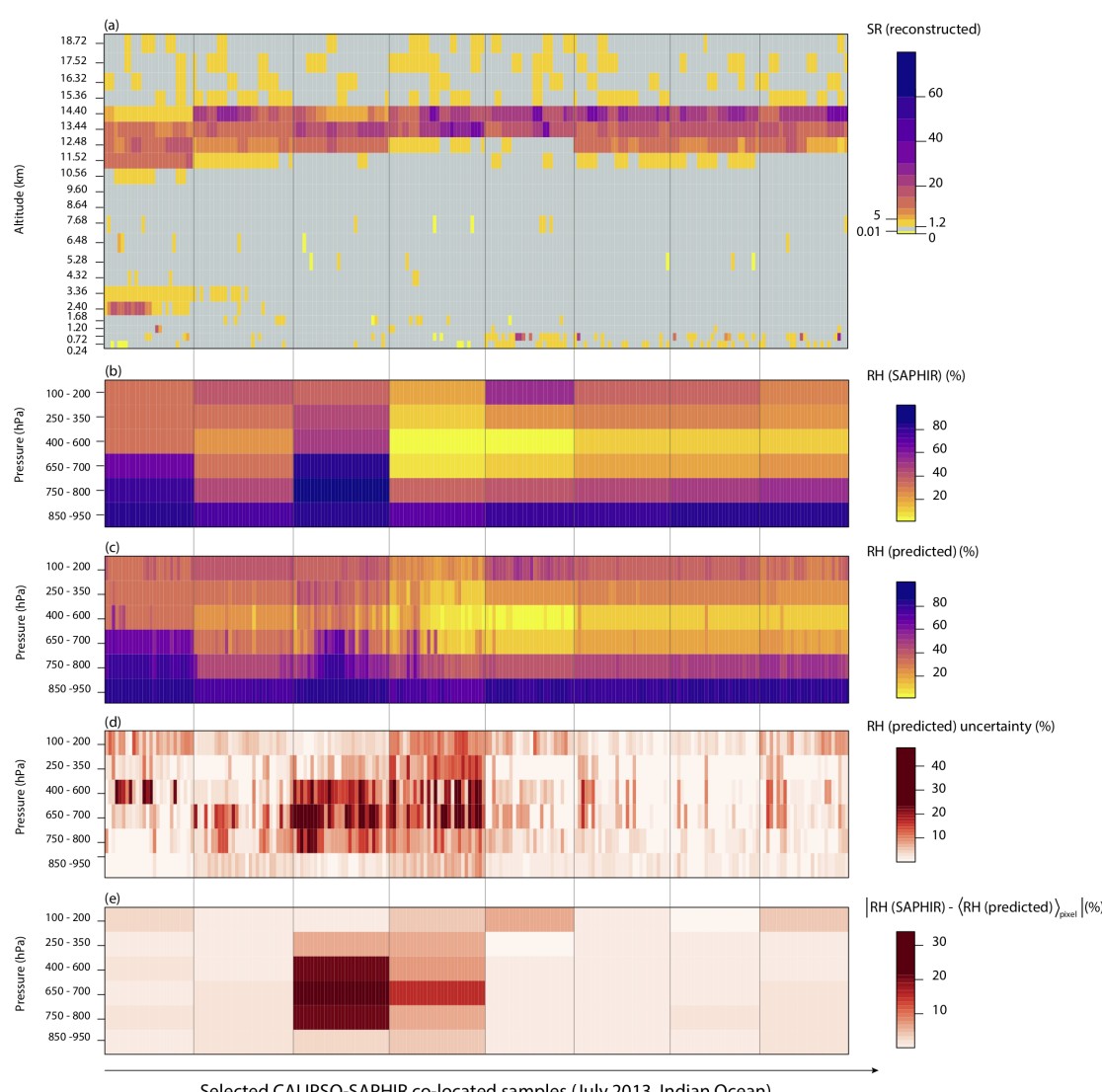

**Figure 8.** (*a*): $SR$ profiles for a selection of ice cloud profiles from CALIPSO in the Indian Ocean, July 2013. The selected cloud profiles correspond to SAPHIR pixels with $n(M) > 25$. The scale is the same as in Fig. 2. (*b*): Co-located layered-$RH$ observations from SAPHIR (mean). (*c*): Predicted layered-$RH$ using the QRF method within the iterative scheme (median). (*d*): as (*c*) but for the interquartile range instead of the median. (*e*): for each layer, absolute differences between the observed $RH$ from SAPHIR and the average over each SAPHIR pixel of the predicted $RH$. The x-axis represents the co-location index.

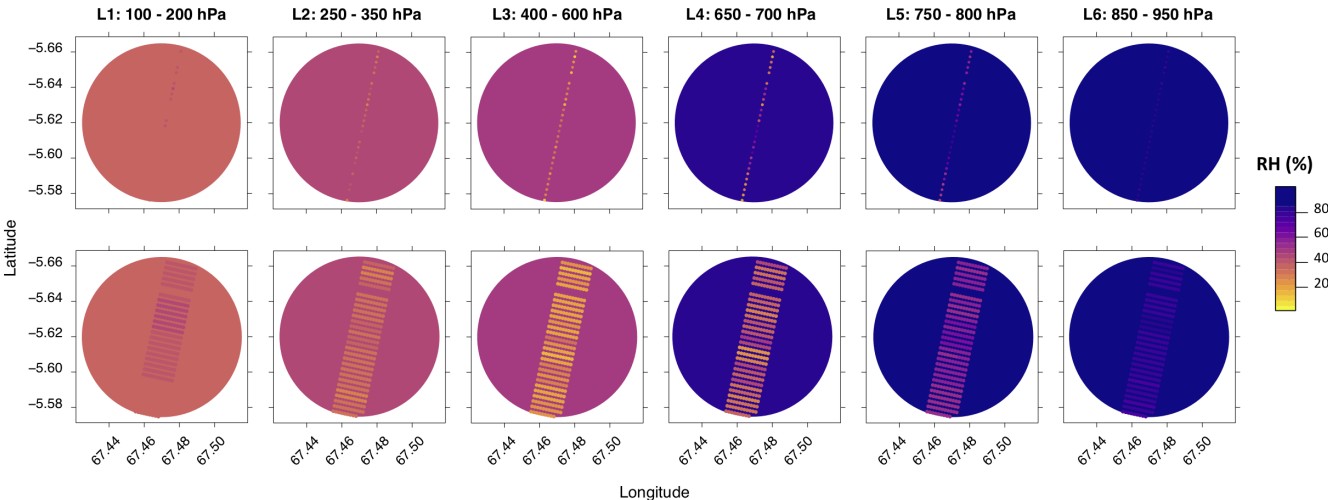

**Figure 9.** Example of predicted $RH$ for a single SAPHIR pixel corresponding to ice cloud profiles using, within the iterative scheme, the QRF method (*top*, median) and the geoadditive model (*bottom*). The disks correspond to the SAPHIR footprints and the dots inside to the $RH$ predictions at CALIPSO resolution. Although CALIOP accumulates data over 330 m along track, here for figure clarity we assumed the profiles to be symmetric and doubled their radius.

| Model | Model type | Spatial Correlation | Prediction type |
| --- | --- | --- | --- |
| RF | Non-parametric | - | Conditional mean |
| QRF | Non-parametric | - | Conditional quantiles |
| GAM | Semi-parametric | - | Conditional mean |
| GAM with GMRF smoother | Semi-parametric | Neighbour structure | Conditional mean |
| Geoadditive | Semi-parametric | Exponential correlation function | Conditional mean |

**Table 1.** Summary of the regression models tested in this study.