# Peer review of "Statistical downscaling of water vapour satellite measurements from profiles of tropical ice clouds"

_Earth System Science Data, 2018_

## Referee Comment (RC1) · Anonymous Referee #1 · 20 Feb 2019

This review is about the article "Statistical downscaling of water vapor satellite measurements from profiles of tropical ice clouds" by G. Carella.

General comment: In the beginning, I was impressed by the approach. Downscaling water vapor from sounder observations could really bring a change to the sounder game. And using microwave sounder information for this is a really promising approach. The article was fluent to read and easy to understand in most parts. The mathematical approaches and the reasoning for modifications are usually well described. However, I also got a little disturbed after I worked through all the fancy mathematics. My main problem: Physically, I do not see a correlation between the SR values from

[Figure]

CALIOP and the layered RH in the atmosphere. So, for me, to a first degree you are applying "magic", covered in – I admit very interesting – mathematical approaches to derive a RH-profile. Ok, do not get me wrong. It is an interesting downscaling and I am still impressed by your approach. I just want to understand some things. I will go through the text chapter by chapter to explain.

1. Introduction: Could you please add, why did you choose CALIOP? For me, there is no physical reason to connect these two pieces of information. CALIOP measures SC at 532 and 1064 nm, which has no connection to water vapor. All information in SC is from cloud particles – mainly ice, in your case. You mention a few articles to infer correlations, but you do not really point out or cite the physical reasoning from there. Please explain, why you think CALIOP is a good choice for water vapor information. I know, your regression models do not need a relationship. But you had your reasons to connect a microwave sounder and a lidar, didn't you?! Because for me, this is not an obvious choice. So please reiterate, why you think this would lead to a physically correct downscaling.

2. Data

Please explain, if the product by Brogniez an official product. Is there an official webpage, source . . . .? Same with CALIOP. Did you use the official product? It seems like, but I want to make sure.

3. Methods

I consider the last sentence in 3.1 is crucial for justification of your technique. You should explain this a little but more, perhaps with more citations from Schroeder's or other papers. It indicates a connection between RH and SR, something which is very important for your approach. But I am not quite sure, how to understand chapter 3.2. You have already a cloud classification from CALIOP (right plot), so why did you prefer your own k-mean method? Both tell you, that clouds above 10km are ice, which is not really big news. You could have just used that value or the CALIOP classification, so

why do you insist to do your analysis based on this extensive k-mean clustering?

Formula 1 in chapter 3.3 is my biggest problem: it assumes RH_l is connected to (SR1, SR2, ... SR_p) via a function. That seems to be the foundation of your idea. But for me, there is none. At least, no physical connection. Is it enough for this approach to find a correlation without reason? I am ok with that, but the results would be of limited use for research (see conclusion). Please re-iterate more here – or at least in the introduction, perhaps based on the articles by Brogniez or Udelhofen-and-Hartmann.

Choice of regression model:

I miss some important information: I understand your limitation to ice clouds. How much data do you use for the regression training with respect to the mission time frame? Do you use specific dates? Do you use the same amount for all regression models (RF,QRF,GMRF1,2,3). Did you make tests with different amounts/dates? Where the results always the same?

Also: Do you deal with the error of the RH-retrieval in RF and QRF? If you look at Figure 2, there are lots of layers with uncertainties > 30%, especially below 500hPa. (Remark: You might want to choose a different color scale there, it is really hard to understand. Everything above 30% is the same color ....). Retrieval tend to get worse closer to the ground. Do you deal with it differently?

I try to understand, why you would chose so many variations of the GAM approach. There is no reason to assume a linear connection between RH and SP, so RF and QRF are quite reasonable to me. But here you suddenly force a linear connection. Is it just for comparison? Because it seems to do bad anyway, when I look at later results. Please re-iterate the reasons for this selection and the two derivatives (GMRF and geoadditive). Are there other options?

Chapter 4: Actually, p.10, line 18-21 is another short alk about a possible physical connection between RH and SR. If you could extend this a little bit more, especially in

the Introduction, then the approach would be much easier to understand. I consider 3 GMF approaches, which have all bad skills, a little bit redundant. I would rather see a third different approach than 3 similar fails. But ok, if you want to keep them, that is fine too.

I also have problems to understand Figure 5. Is the predicted from RF? And is the observed RH the one from 10x10km SAPHIR? The description in the text is very short and confusing. Please excplain more here: source of predicted, source of observed. Why would you then have such a bad correlation for L6? Please explain this plot in more detail, it seems it is your only source of verification for your approach. Most people would prefer an independent source (radiosonde, airplane observation, ….), but I guess you don't have enough data for this in the Indian ocean. So, you have to convince the reader about the "success" of your approach with this plot. Honestly, I didn't get convinced, you didn't write enough.

Chapter 5:

At the moment, I am questioning some bullet points in your conclusion. I am not quite convinced that your data can help "study . . . small scale water processes" or "evaluate . . . water vapor interactions". You need to convince me, that you have a physical foundation, not just correlated sorting. On the other side, I agree that you can always "evaluate small scale inhomogeneities" in reanalysis or "guide parameterizations". Models need to know the behavior of parameterizations on smaller scales, so you might be very helpful to find out scale breaks on scales around 100 m. I also think, you should talk a little bit more about the extension to other clouds. It sounds interesting, but based on your requirements (homogeneity, strong SP signal), you might be in trouble. If you could talk more about future possibilities and obstacles, it would be a better selling point for this article. But that is more my opinion. . .

Minor comments, found during reading: p. 2, line 3: Should be "state-of-the-art" p. 2, line 31: I am not quite sure, what "space clouds" are . . .. p. 5, line 10: should be "nadir"

p. 6, line 1 : is the "1" necessary here? p. 10, line 2-4: this sentence is hard to read with all the comma and brackets. I would propose to redo it a little bit. p. 12, line 16: should be "CALIPSO"

Still, I consider it a very interesting article. My Regards.

---

## Referee Comment (RC2) · Anonymous Referee #2 · 7 Mar 2019

Review of manuscript ESSD-2018-138 : "Statistical downscaling of water vapour satellite measurements from profiles of tropical ice clouds"

This manuscript deals with fine-scale water vapor retrievals derived from a combination of satellite instruments using a downscaling technique. The paper is well structured and should be published after some more details and explanations given to help the readers understanding the methodology and its limitations. My main concerns are as follows:

- The need for fine scale observations of the vertical structure of water vapor is clear and well justified. But I probably missed a major thing reading the manuscript: from

[Figure]

Figures 8 and 9, it seems to be more of a horizontal downscaling of the SAPHIR RH product than a vertical downscaling of it. Please clarify this either in the introduction of in the results. As I said, I might have missed something but I may not be the only one when reading your work.

- The results shown on Figure (8e) indicate that the differences between the predicted RH and the RH estimated from SAPHIR can be quite large. It seems that Figure (8e) is not commented at all in the text but it needs explanations. Can the differences be explained by representativeness errors between the CALIOP lidar and the SAPHIR radiometer?

- The results shown in Figure 9 where RH estimated from SAPHIR and the predicted RH are on the top each other seem to indicate there is a bias between the two, especially in the lower layers. Could you please comment on this? In the paragraph page 11 where these results are presented, there is a comment on the variance of the predicted RH but not on its bias.

- In the Data section on CALIPSO data, it is shortly explained that the noise on the profiles has been reduced using a Principal Component Analysis to keep only 90% of the variance. Why 90%? Would have the results fundamentally changed if you hadn't done this filtering?

Minor comments:

- Figure 1 shows a case of January 2nd in 2017 but the rest of the examples are for July 2013. Is there a way you could update Figure 1 to show the same meteorological situation all along the manuscript?

- Page 2, line 29 : "These detailed profiles are observed all over the globe" => Isn't SAPHIR observing the Tropics only? Please correct this sentence.

- Page 5, line 8, "3.1 SAPHIR-CALIPSO co-location" => The period of the study is not mentioned here but that we be good to know at this stage and not only later in Section

3.2

- Page 5, line 23 : "recontructed" => reconstructedÂă

- Page 10, line 32 : "on the distance from the cost" => coast

---

## Author Comment (AC1) · 1 May 2019

1. Introduction

Could you please add, why did you choose CALIOP? For me, there is no physical reason to connect these two pieces of information. CALIOP measures SC at 532 and 1064 nm, which has no connection to water vapour. All information in SC is from cloud particles – mainly ice, in your case. You mention a few articles to infer correlations, but you do not really point out or cite the physical reasoning from there. Please explain, why you think CALIOP is a good choice for water vapour information. I know, your regression models do not need a relationship. But you had your reasons to connect a microwave sounder and a lidar, didn't you?! Because for me, this is not an obvious choice. So please reiterate, why you think this would lead to a physically correct downscaling.

As clearly pointed here by the reviewer, from a purely remote sensing point of view, there is indeed no connection between the backscatter at 532 nm and 1064 nm and the water vapour concentration. However, numerous studies have highlighted that there are relationships between the presence of upper tropospheric ice clouds and the surrounding moisture. For instance, Jensen et al. (1996, GRL; 2001, JGR) and Rosenfield et al. (1998, GRL) showed that the formation of thin cirrus is associated to the dehydration of the upper troposphere, essential for their maintenance. Luo & Rossow (2004, J. Clim.) as well as Soden (2004, GRL) and Chung et al (2007, ACP) related the life cycle of cirrus anvils to an increase in the moisture content of their close environment, via detrainment. This was also highlighted by Eguchi & Shiotani (2004, JGR). Moreover, Martins et al (JGR, 2011) examined the link between optically thin ice cloud observed by CALIPSO and the collocated upper troposphere water vapour from the Microwave Limb Sounder (MLS) on the Aura platform and found that cirrus cloud detections in the upper troposphere are correlated with a significant increase in the observed upper tropospheric water vapour concentrations compared to the average. The link between ice clouds and water vapour was also examined in Hoareau et al. (ASL, 2016) using ground base lidar observations collected at la Réunion island in the tropics. On the other hand, as for example explained in Cesana and Chepfer (2013, JGRA), the lidar scattering ratio (SR) also depends on the amount of condensed water (and therefore a mix of concentration, size and shape of ice crystals in the atmosphere) and increases with the amount of condensed ice in the atmosphere (only when the cloud optical depth <3, which is the case for most ice clouds). Therefore, given that both water vapour and the lidar intensity are linked to ice crystals concentration, size and shape, we expect some correlation between the measured RH by SAPHIR and the SR signal from CALIPSO. This being said, the retrieval of RH from the microwave sounder is not biased by the presence of ice particles. Therefore, in the following, we assumed that the retrieved RH from SAPHIR can indeed be reasonably predicted given CALIOP measurements of ice clouds.

We have added some text in the manuscript (page 3, lines 10-28) to clarify this point.

2. Data

Please explain, if the product by Brogniez an official product. Is there an official web-page, source . . ..? Same with CALIOP. Did you use the official product? It seems like, but I want to make sure.

- The RH profiles from SAPHIR used here are depicted in Sivira et al. (2015, AMT) and in Brogniez et al. (2016, JAOT) and this is indeed the official product. It can be downloaded, after registration, from the French ground segment of the Megha-Tropiques data:

  http://www.icare.univ-lille1.fr/mt/

- The scattering ratio profiles from the lidar CALIPSO used in this study are from the GCM-Oriented Cloud-Aerosol Lidar and Infrared Pathfinder Satellite Observations (CALIPSO) Cloud Product (CALIPSO-GOCCP, Chepfer et al., 2010)). GOCCP product is part of the CFMIP-OBS database (Cloud Feedback Model Intercompariosn Program). It has been compared to the NASA CALIPSO product in Cesana et al. 2016 (JGR) and Chepfer et al. 2013 (JAOT). Data can be downloaded from:

  http://climserv.ipsl.polytechnique.fr/cfmip-obs/Calipso_goccp.html

**3. Methods**

3.1. I consider the last sentence in 3.1 is crucial for justification of your technique. You should explain this a little but more, perhaps with more citations from Schroeder's or other papers. It indicates a connection between RH and SR, something which is very important for your approach.

This comment on the connection between RH and SR is similar to what was raised in the comment #1 (Introduction). Please refer to that comment for further justification.

We have also deleted the reference to Schröder et al. (2017), since in that study the authors are not mentioning the fact that the correlation between ice clouds and Upper Tropospheric Humidity is large.

3.2. But I am not quite sure, how to understand chapter 3.2. You have already a cloud classification from CALIOP (right plot), so why did you prefer your own k-mean method? Both tell you, that clouds above 10km are ice, which is not really big news. You could have just used that value or the CALIOP classification, so why do you insist to do your analysis based on this extensive k-mean clustering?

By averaging the SR profiles above the boundary layer to a 1 km resolution with the aim of reducing the noise and the amount of missing data, we also had to apply the same averaging procedure to the cloud phase flag profiles in order to maintain a coherence between the SR profiles used in the regression model and the corresponding cluster. Because of the "mixed" flags resulting from this averaging procedure, the statistically-based clustering method was preferred since it encompasses the problem of giving a physical interpretation to "mixed" profiles (c.f. Fig. 3c). Moreover, in this way, the downscaling method described in this study can

be more easily generalized, without having to worry about the physical interpretation of the clusters.

We have added some text in the manuscript (page 6, lines 19-26) to clarify this point.

3.3 Formula 1 in chapter 3.3 is my biggest problem: it assumes RH_l is connected to (SR1, SR2, . . . SR_p) via a function. That seems to be the foundation of your idea. But for me, there is none. At least, no physical connection. Is it enough for this approach to find a correlation without reason? I am ok with that, but the results would be of limited use for research (see conclusion). Please re-iterate more here – or at least in the introduction, perhaps based on the articles by Brogniez or Udelhofen-and-Hartmann.

Once again, this comment discusses the connection between RH and SR is similar to what was raised in previous comments, 1 and 3.1. Considering the expected physical correlation between RH and condensed ice (crystal concentration, shape and size), as well as the published papers that have highlighted the relationships between upper tropospheric moisture and ice clouds, we think that we have sound reasons to link, via a function, RH and SR.

See also the detailed response to comment #1.

3.4. Choice of regression model: I miss some important information: I understand your limitation to ice clouds. How much data do you use for the regression training with respect to the mission time frame? Do you use specific dates? Do you use the same amount for all regression models (RF,QRF,GMRF1,2,3). Did you make tests with different amounts/dates? Where the results always the same?

As stated in the manuscript (c.f. for example Figure 7) we tested the method using different time periods (July and January 2013) and different ocean basins (Indian and Pacific). All methods were trained on the same data, in order to allow for a fair comparison.

3.5. Do you deal with the error of the RH-retrieval in RF and QRF? If you look at Figure 2, there are lots of layers with uncertainties > 30%, especially below 500hPa. (Remark: You might want to choose a different color scale there, it is really hard to understand. Everything above 30% is the same color . . ..). Retrieval tend to get worse closer to the ground. Do you deal with it differently?

In this study, we did not account for errors in the RH retrieval (we used the mean of the RH distribution from the retrieval algorithm) but this point can be further developed in future studies.

Larger uncertainties in the retrieved RH are expected at lower altitude because of the distribution of the sounding channels of the radiometer and because of their bandwidth (Clain et al., 2015 JAOT). The latter is narrow (0.2 GHz) for the central channels of the 183.31 GHz absorption line, which translates into a low uncertainty for the upper tropospheric estimates, and it stretches (2 GHz) for the channels located in the wings of the line, implying a larger uncertainty for the retrieval. Overall, RH measurements

with a standard deviation larger than 30% might be considered very uncertain (which explains the chosen colour scale).

We have added some text in the manuscript (page 6, lines 7-14) to highlight this point.

3.6. I try to understand, why you would chose so many variations of the GAM approach. There is no reason to assume a linear connection between RH and SP, so RF and QRF are quite reasonable to me. But here you suddenly force a linear connection. Is it just for comparison? Because it seems to do bad anyway, when I look at later results. Please re-iterate the reasons for this selection and the two derivatives (GMRF and geoaddtive). Are there other options? I consider 3 GMF approaches, which have all bad skills, a little bit redundant. I would rather see a third different approach than 3 similar fails. But ok, if you want to keep them, that is fine too.

First, note that, as explained in section 3.3.1, a "Generalized Additive Model" (GAM) is not a linear model. Moreover, compared to tree-based models, GAMs, in addition to offering the advantage of interpretability of the model coefficients, also allow the direct incorporation of a spatial correlation structure. Although this turns out not to be important for this particular downscaling application (as shown in section 4), we believe it is still useful to include the description and the results of these models in the manuscript for potential applications of the method to different data and problems.

3.7. Chapter 4: Actually, p.10, line 18-21 is another short talk about a possible physical connection between RH and SR. If you could extend this a little bit more, especially in the Introduction, then the approach would be much easier to understand.

We have extended the discussion on the link between RH and SR, as suggested also by the previous comments.

3.8. I also have problems to understand Figure 5. Is the predicted from RF? And is the observed RH the one from 10x10km SAPHIR? The description in the text is very short and confusing. Please explain more here: source of predicted, source of observed. Why would you then have such a bad correlation for L6? Please explain this plot in more detail, it seems it is your only source of verification for your approach. Most people would prefer an independent source (radiosonde, airplane observation, ....), but I guess you don't have enough data for this in the Indian ocean. So, you have to convince the reader about the "success" of your approach with this plot. Honestly, I didn't get convinced, you didn't write enough.

Figure 5 shows the median of the distribution of the predicted RH for each vertical layer using the Quantile Random Forest method vs. the RH observed by SAPHIR (at 10x10 km resolution). Here the predictions are the results of the 5-fold cross validation procedure, and are therefore derived from a model trained on an independent part of the data set. Although a comparison with other sources of RH data, as for example those cited by the reviewer, could be interesting, it will not necessarily be a validation of the results of our model. In fact, apart from the difficulty of finding a statistically significant sample of radiosondes or airplane observations co-located in space and time with CALIPSO measurements, these sources are characterized by different spatial resolutions from lidar data, which makes the comparison not  straightforward.

We have added some text in the manuscript (page 11, lines 18-31 and page 12, lines 1-6) to clarify what Figure 5 is.

4. Chapter 5

At the moment, I am questioning some bullet points in your conclusion. I am not quite convinced that your data can help "study . . . small scale water processes" or "evaluate . . . water vapour interactions". You need to convince me, that you have a physical foundation, not just correlated sorting. On the other side, I agree that you can always "evaluate small scale inhomogeneities" in reanalysis or "guide parameterizations". Models need to know the behaviour of parameterizations on smaller scales, so you might be very helpful to find out scale breaks on scales around 100 m. I also think, you should talk a little bit more about the extension to other clouds. It sounds interesting, but based on your requirements (homogeneity, strong SP signal), you might be in trouble. If you could talk more about future possibilities and obstacles, it would be a better selling point for this article. But that is more my opinion. . .

As stated in comment #1, water vapour and the lidar intensity both are linked to ice crystal concentration and size (and shape). This represents the physical foundation of our method and implies that the correlation observed between SAPHIR RH and CALISPO SR emerges from a physical relationship, which indeed mean that the downscaled profiles can then be used to help to study small scale water cycle processes.

We do not agree that our method requires homogeneity or a strong SR signal. On the other hand, it is true that liquid clouds could present additional challenges in the implementation of the method. In fact, while SAPHIR is not able to retrieve the RH profile in the case of heavy precipitation, which implies that the majority of ice clouds co-located with SAPHIR measurements are non-precipitating, this is not true for light precipitating clouds, which typically correspond to low-level liquid clouds only. Therefore, for liquid clouds, including the radar reflectivity as measured by the radar CloudSat, which is indicative of the intensity of rainfall, might increase the model explanatory power.

We have added some text in the manuscript (page 14, lines 25-28) to clarify this last point.

5. Minor comments, found during reading:

- p. 2, line 3: Should be "state-of-the-art"

    Changed.

- p. 2, line 31: I am not quite sure, what "space clouds" are . . ..

    Changed.

- p. 5, line 10: should be "nadir"

    Changed.

- p. 6, line 1 : is the "1" necessary here?

Table numbering is required by the journal.

- p. 10, line 2-4: this sentence is hard to read with all the comma and brackets. I would propose to redo it a little bit.

Changed.

- p. 12, line 16: should be "CALIPSO"

Changed.

**Anonymous Referee #2**

1. The need for fine scale observations of the vertical structure of water vapour is clear and well justified. But I probably missed a major thing reading the manuscript: from Figures 8 and 9, it seems to be more of a horizontal downscaling of the SAPHIR RH product than a vertical downscaling of it. Please clarify this either in the introduction of in the results. As I said, I might have missed something but I may not be the only one when reading your work.

The reviewer is correct in pointing out that the focus of this paper is the horizontal downscaling of RH profiles at the resolution of cloud measurements. The main interest of this study is in fact to test a statistical approach to overcome the barrier of the coarse footprint size of the radiometer, which implies that small-scale heterogeneities in the RH field are missed.

The coarse vertical resolution is also critical (not less), especially in cases where there are strong vertical gradients of moisture. For instance, at the top of the atmospheric boundary layer over the oceans in regions of shallow clouds (stratocumulus or cumulus) the boundary layer can be really moist, near saturation, whereas the free troposphere above can be extremely dry. Similarly, at the Upper Troposphere/lower Stratosphere boundary, the moisture is really low and this is critical for the ozone budget. However, these are two different topics that could indeed be tackled with similar approaches, but require different sets of proxies.

We have added some text in the manuscript (page 4, lines 13-20) to clarify this point.

2. The results shown on Figure (8e) indicate that the differences between the predicted RH and the RH estimated from SAPHIR can be quite large. It seems that Figure (8e) is not commented at all in the text but it needs explanations. Can the differences be explained by representativeness errors between the CALIOP lidar and the SAPHIR radiometer?

Differences between the downscaled and the observed RH observations will be larger when the RH field is characterized by finer-scale heterogeneities deriving from finer-scale processes, as for instance Figure 8e seems to suggest for some of the profiles. However, these differences are expected since with the method presented here the predicted relative humidity structure incorporates the higher-resolution variability from

cloud profiles. On the other hand, as shown both in Fig. 4, 5 and 7, the downscaling model is able to successfully explain the coarse-scale RH observations from the finer-scale SR measurements and the overall bias is low, which gives us confidence in the predictions.

We have added some text in the manuscript (page 12, lines 29-32 and page 13, lines 1-2) discussing Fig. 8e.

3. The results shown in Figure 9 where RH estimated from SAPHIR and the predicted RH are on the top each other seem to indicate there is a bias between the two, especially in the lower layers. Could you please comment on this? In the paragraph page 11 where these results are presented, there is a comment on the variance of the predicted RH but not on its bias.

The model bias is overall low, as discussed in the previous point. On the other hand, what Fig. 9 is showing is that the variations explained by the spatial smoothing are negligible, and that the SR predictors alone explain the largest component of the variance in the RH field. In other words, once the effect of the SR predictors is taken into account, the residuals (i.e. the difference between the observed and the predicted RH) do not show spatial autocorrelation.

We have added some text in the manuscript (page 13, lines 8-10) discussing Fig. 9.

4. In the Data section on CALIPSO data, it is shortly explained that the noise on the profiles has been reduced using a Principal Component Analysis to keep only 90% of the variance. Why 90%? Would have the results fundamentally changed if you hadn't done this filtering?

Before clustering of the SR profiles and for clustering only, we decided to keep the number of PC components explaining the 90% of the variance (resulting in 19 retained PCs) as little variance was gained by retaining additional components. However, the results of the study did not change fundamentally when no noise reduction was applied prior to clustering (not shown). To improve clarity, we moved the description of the PCA analysis to section 3.2 and refer to the textbook of von Storch and Zwiers (1999).

5. Minor comments:

- Figure 1 shows a case of January 2nd in 2017 but the rest of the examples are for July 2013. Is there a way you could update Figure 1 to show the same meteorological situation all along the manuscript?

  We left Figure 1 as it is because here it is only used as a schematic representation of the method and is not intended to give any physical insight.

- Page 2, line 29 : "These detailed profiles are observed all over the globe" => Isn't SAPHIR observing the Tropics only? Please correct this sentence.

  Changed.

- Page 5, line 8, "3.1 SAPHIR-CALIPSO co-location" => The period of the study is not mentioned here but that we be good to know at this stage and not only later in Section 3.2

  The layer-averaged RH profiles from SAPHIR from Brogniez et al. (2016) are available for the period October 2011 – present, while CALIPSO-GOCCP product is available between June 2006 and December 2018. This has been clarified adding some text in section 2.1 (page 4, line 21-22) and section 2.2 (page 4, line 31 and page 5, line 1) respectively.

- Page 5, line 23 : "recontructed" => reconstructed˘

  Changed.

- Page 10, line 32 : "on the distance from the cost" => coast

  Changed.

---

## Referee Report (RR1)

Review of the manuscript "Statistical downscaling of water vapour satellite measurements from profiles of tropical ice clouds" by Giulia Carella, Mathieu Vrac, Hélène Brogniez, Pascal Yiou, and Hélène Chepfer.

In the manuscript, a novel approach for approximating the relative humidity (RH) in the middle atmosphere from the observations of scattering ratio (SR) of the CALIPSO satellite instrument is presented. Several machine learning models are exploited as the primary approach along with additional iterative corrections aimed to preserve the average estimated RH to be close to observed by SAPHIR instrument. Preprocessing of source data includes spatial co-location of SAPHIR and CALIPSO sounding locations; averaging of characteristics of cloud profiles above 2km; filtering of CALIPSO observations related to ICE and ICE-MIX clouds based on k-means clustering results and additional filtering.

Overall I find the idea and the approach of the study to be very promising. The models exploited in the study are described in a clear manner, without over-complications or obscured details. I find the English of the manuscript a bit knotty, but acceptable.

General comment:
The main concern about the presented study is the unclear general setup of the problem. It is clear that the covariates for a model (one of five regression models) are SR values preprocessed in a way described in Section 2.2. It is also clear that the target values of a model are RH at the points at fine spatial resolution colocated with CALIPSO sounding events. However, it is unclear which values are the ground truth for these target values. Are they just the value of SAPHIR pixel RH? Then it should be mentioned clearly.
As far as I understand the approach of the authors, the general setup of the problem is as follows:
SR (at p=21 height layers) are the covariates for a model. SR are defined at fine spatial (horizontal) resolution. $RH_l$ (at 6 pressure levels) are target values for a model. $RH_l$ are defined at coarse spatial resolution. A model is fitted to estimate $RH_l$ at each point of CALIPSO sounding based on SR values of these points. There is also a step for preserving the so-called "mass balance." And there is no "ground truth" for RH values at the fine spatial resolution other than coarse SAPHIR data.
Following this setup, the authors fitted the models they presented in Section 3.3.1. As a result, the authors obtained RH at a fine (horizontal) resolution with some variability within each SAPHIR pixel. However, there is no reason to claim this variability to be downscaled structure of RH within SAPHIR pixel. The uncertainty of the $RH_l$ value estimated be a model within a SAPHIR pixel is explained by two factors: RH variability itself and errors of a model. The authors did not mention in any way how did they determine the part of the uncertainty that is related to RH variability. And, as far as I understand the setup of the presented study, there is no reason to claim the variability of $RH_l$ value estimated by a model to be a fine structure of RH. It is especially arguable in case of non-parametric models like RF or QRF.
According to the above, the naming of the study is confusing and misleading. I would totally agree with the title mentioning "approximation" or "regression" of RH based on CALIPSO profiles. However, the "downscaling" naming seems incorrect in this case.
Considering those mentioned above, the discussion about the downscaling results is also unconvincing (p.12 L27 - p.13 L12). Taking the design of the study into account, one cannot claim to be able to obtain variations of RH within SAPHIR pixels.

The conclusions related to the downscaling feature of the presented approach are incorrect in this case as well. The results of a model can be considered as an estimate of RH with some uncertainty, but cannot be considered as a downscaling product with estimates of inhomogeneities.

Section 3.2:
The classification of the SR profiles based on clustering techniques does not seem reliable. Since clustering itself is a tool for explorative analysis rather than classification, one cannot claim the clustering result to be the answer for the question "which profiles correspond to ice clouds and which do not." It is not clear how do authors checked if the profiles of the cluster 1 indeed correspond to ICE and ICE-MIX observations (p. 7, L3-9). As it seems at the current state of the manuscript, the similarity of clustering made with k-means with the observations characterized as ICE and ICE-MIX was assessed visually. If that is the case, it does not seem like a convincing way of detection of ice clouds. It is also not clear why the authors did not use cloud phase flags instead of this additional clustering approach.

Section 4:
The results described in Section 4 (p. 12, L1-2) are inconsistent with Figure 5 in terms of the values of the coefficient of determination. $R^2$ for L6 mentioned to be ~0.3 in the text of the manuscript; meanwhile, it is 0.4 in the Fig.5. Also, for the layers L1-L5 the values of $R^2$ claimed to be >=0.7, and as presented in the figure, are precisely 0.7. While formally, it is not a mistake; it seems like a mistyping in the figure.

Fig.7:
The description of results shown in Fig.7 is not clear. It is unclear what is the statement to which "Similar results can be found for different choices of the number of clusters..." is related (p.12, L12-15)

Fig.3:
The horizontal axis meaning is not clear enough. It should be labeled with some kind of "cluster number" of another informative label.

---

## Author Response (AR2)

**Reviewer 1:**

Ok, I really have a problem with the answer to my first question. The question was "why did you choose CALIOP?" And the answer is "there is indeed no connection between backscatter at 532 nm and 1064 nm and the water vapour concentration".

This is followed by a variety of interesting articles as a proof for this connection. The problem: none of them gives proof any good connection between water vapor and a lidar.
Actually, they proof more the opposite.

For example, Jansen (1996, 2001) mainly explain the dehydration during ice cloud formation. But this paper also shows, that you can easily have variations of 100-160% humidity without cloud formation. Such a change in humidity would be visible in a microwave system, but I doubt there would be a signal in the lidar. Because lidar reacts only to scattering, means ice particles.
The other thing that becomes obvious, for example in Jensen et al., 2001: Due to wind advection you can have large changes in ice water content, even if the water vapor mixing ratio is constant (see for example Figure 5 of the reference above). The result: your lidar actually reacts differently, but microwave does not.
Luo & Rossow (2004) is not really relevant here, if you ask me. They looked more into life cycles. And the left part of Fig 14 in that paper proofs actually the opposite again: IWP is changing over time, extra water vapor is changing. So, your Lidar would have a constant signal, WM would see a variation. You could argue, that you observe only steady state (the right part of the Figure). But CALIOP is a polar orbiter, so you would have to justify that somehow.
Eguchi & Shiotani (2004) are focusing scales of Rossby- and Kelvin-waves. Your downscaling is on a kilometer scale. If you look at Fig. 4 and 5 there, then you can see that there is no exact match of cirrus frequency and water vapor mixing ratio. Figure 7 makes it even more obvious. And that is on a 1-degree scale. Is there a reason that you assume it works on a km scale?
Your best argument is Cesana and Chepfer (2013), where you make the case that there is an "indirect correlation" between RH and ice particle (shape, size,…)
However, it is also another good example for the opposite argument: when looking at Figure 13. water vapor and cloud fraction do not overlap

Ok, I am not going through all the other articles, the outcome is similar: None of them show a perfect correlation between water vapor and cloud properties (fraction, particle size, shape, ...).

Don't get me wrong. I think you made a good argument with Cesana and Chepfer. But you have to exploit it and explain it. You have to point out that water vapor is not measured directly by CALIOP. You have to explain that cloud properties (fraction, particle size, thickness, phase) will be the dominant signal in this approach.
And I propose, you have to state this very obvious and in the beginning. It makes sure that people will not use your "water vapour product" to proof a correlation with cloud properties, for example for CALIOP. (People might still do it afterwards, then they cannot be helped.)

My proposal: Include a separate chapter after the introduction. Explain your approach and the uncertainties connected with it in more detail. Explain, that water vapor is not measured directly by the lidar. Explain your assumption more detailed: small scale cloud variations in the cloud (particle size, phase, …) are the result - or lead to - small scale water vapor variations. Cause and effect are a matter of debate here and you can actually find citations for both sides, if you look into it. But they are more in the direction of cloud microphysics papers, not in the large scale dynamics, which you cited before. And last but not least: point out, when this product is not useful (e.g. doing a comparison of water vapour and lidar cloud properties)
You spend a lot of time later to describe your mathematical approach, so one little chapter about the physical idea would add a lot of clarification for possible interested users.

The rest of the article was answered and clarified satisfactory. So, as soon as the "physical interpretation" is clarified, I have no problem with the publication later.

We would like first to thank the reviewer for this very constructive feedback, and we agree that a full paragraph dedicated to the physical background is necessary. We have thus added a section (page 5 line 11 – page 6, line 18) to provide such background, as well as the limits of the approach.

3. Physical approach and related limitations

[revised manuscript text omitted]

**Reviewer 2:**

The primary goal of this paper is to downscale vertical profiles of relative humidity (RH) from the SAPHIR instrument from the native 10-km horizontal resolution to 90-m horizontal resolution. This approach relies on a profile of scattering ratio from the CALIOP lidar that is collocated to SAPHIR then clusters containing tropical ice cloud data only are used to infer ice cloud features within vertical profiles of RH. A statistical methodology is then used to obtain downscaled profiles of RH at very high spatial resolution (90 m).

The science justification for wanting to derive small scale RH structures is strong and additional structure in RH at small scales from satellite data would be very useful for atmospheric research. With that said, I found this paper very difficult to understand at times and why the authors made particular methodological choices without explaining them makes the presentation confusing.

We have worked on the manuscript to make it clearer, thanks to the next comments and to the other referees' comments.

1. Some of the figures are problematic: figs 2 and 8 do not have x-axis scales

The x-axis scales in these figures represents the index of the co-located CALIPSO and SAPHIR samples. It is thus neither a time nor a spatial axis.
We labelled the x-axis as the index of the co-located samples to make it clear.

2. fig 5 uses a light shade of yellow that underplays the degree of scatter

We added a contour line to highlight each data point in the plot.

3. and fig 9 needs lots of work to improve and virtually no discussion on the results is included. There are many studies in the literature relevant to this topic that are ignored and would be very helpful for informing the author's methodologies and validation strategies. Below are some general comments that I hope the authors would consider to improve the clarity of their potentially promising, yet still immature results. There was no "proof of concept" with independent validation or even something as simple as a statistical summary/ histograms of the small scale features in RH. Figure 9 is really confusing and very little effort is made to explain the final product of downscaled RH. Why not show some summary statistics: mean values, standard deviations, perhaps skewness varies with height? There is enough information in the literature to develop a fuller story to end the paper and make comparisons. Some basic summary statistics are mandatory in this reviewer's opinion.

As explained in the newly added section 4.3.3. (page 11, line 23 – page 12 line 28), in the case presented in this study, no RH observations at the horizontal resolution of the cloud measurements (or higher) are available such that, when co-located with CALIPSO data, provide a large enough training or even testing set for the regression model. The fidelity of the method can therefore only be evaluated at the level of the available coarse observations, while a 'mass-balance' correction can be applied to ensure the best possible consistency with the original measured values. Although we agree that it would be interesting to derive some basic statistics to compare the original and the downscaled product, we also think that, in order to derive meaningful statistics, it is necessary to have a much larger sample, which could be obtained by applying the method on all years, locations and cloud types of available data. This is in fact the object of a future study, while the current manuscript is intended to present the method only.

4. There is no discussion on how one would interpret a 1 km vertical x 90 m horizontal observation (90 m in one or two dimensions? what about the second dimension across the lidar track? also the small scale patches in this figure looks coarser than 90 m. how do we interpret a highly asymmetric 3d product?)

Figure 9 aims to show the prediction of RH on the side of CALIPSO track, where no cloud data are available, and assuming that each SR profile is also representative of the cloud distribution along the direction orthogonal to CALIPSO track within a distance of 1 km.

CALIOP accumulates data over 330 m along track with a beam of 90 m at the Earth's surface. This means that, although the individual laser beam has a field of view of 90 x 90 m at the Earth's surface, because the satellite moves, the data are accumulated over 330 m in order to reduce the signal-to-noise ratio. Indeed, for figure clarity, in Fig. 9 we assumed the profiles to be symmetric and doubled their radius.

Following these remarks, we modified the caption to Fig. 9, which now reads:

"Example of predicted RH for a single SAPHIR pixel corresponding to ice cloud profiles using, within the iterative scheme, the QRF method (*top*, median) and the geoadditive model (*bottom*). The disks correspond to the SAPHIR footprints and the dots inside to the RH predictions at CALIPSO resolution. Although CALIOP accumulates data over 330 m along track, here for figure clarity we assumed the profiles to be symmetric and doubled their radius."

5. The authors ignored the literature on atmospheric sounding besides a cursory mention of Microwave Limb Sounder data. There was no mention of infrared hyper spectral sounders AIRS, CrIS, and IASI which are quite useful for sounding RH information within thinner cirrus clouds, as they have higher vertical resolution than microwave SAPHIR 183 GHz channels. There are many publications that exploit RH in clear sky and within ice/cirrus clouds and none are mentioned.

We reckon that there are some publications that exploit RH in the presence of cirrus clouds, and that the infrared hyperspectral sounders that are mentioned (AIRS, CrIS, IASI) are very useful to look at situations with optically thin ice clouds. The issue with those instruments are the same as with SAPHIR: the horizontal resolution is coarse.
However, the scope of the present paper is not to focus on a type of cloud, but to develop a method that can be applied to any cloud, be it optically thin or thick (where the IR sounders are not able to get the RH). And we used SAPHIR data as "first-guesses" of RH profiles but this method can be, in theory, applied to other instruments. We do not think that there is a need to recall all the possibilities for RH sounding from space-borne platforms: we believe that such historical recall would definitely weigh down the message of the paper and then miss its scope.

6. The distributions of RHi are closer to 100% with lower standard deviations in cirrus compared to clear skies in proximity to cirrus, with much more variable RHi and more frequent supersaturation. To add to this, there are also many papers on in situ aircraft observations of RHi distributions in cirrus/clear sky and the scale dependence of whether temperature or water vapor controls the RHi distributions is inferred from HIPPO aircraft observations.

No separate error propagation as a function of scale for temperature and specific humidity is attempted.

As passive instruments (SAPHIR, here, but it would be the same for another passive sensor) measure a mixture between the concentration of the absorber (water vapor) and temperature (through the Planck's function), RH is the most direct quantity measured. Making the separation between temperature and specific humidity would require a priori assumptions on one of these two variables, as discussed in section 2.1 in Brogniez et al. 2013. Therefore, no separation can be made without an additional temperature or specific humidity profile. That is why we cannot compute the suggested propagation of error.

We modified section 2.1 (page 4, lines 7-10), which is dedicated to the description of SAPHR data, in order to state that working in the RH space is straightforward while a translation into temperature or specific humidity cannot be performed a priori. The modified text in section 2.1 now reads:

"In this line of strong absorption of radiation by water vapor, the measured radiation is affected both by the absorber amount (the water vapor) and the thermal structure, making the retrieval of RH more straightforward and less dependent on a priori temperature or absolute humidity data (Brogniez et al., 2013)."

7. There are no cited papers on power law relationships that highlight regime and height and cloud/clear sky dependence of variability as a function of scale, and the existence of scale breaks in the exponents and how that impacts the small scale RH.

Indeed, we are aware that variations of relative humidity can be expressed in terms of power laws. This was discussed for instance in Raymond (2000). While such formulation is used in an analytical framework, such as for sub-grid parametrisation in climate or regional models, we don't see the point of adding such discussion in the present study.

Raymond, W.H.: Estimating Moisture Profiles Using a Modified Power Law. J. Appl. Meteor., 39, 1059–1070, doi : 10.1175/1520-0450(2000)039<1059:EMPUAM>2.0.CO;2, 2000.

8. How does the statistical methodology used in this paper deal with known scale dependent variability in temperature, specific humidity, and cloud geometrical and microphysical properties?

As for the previous comment, we think that this scale-related remark does not fall in the scope of the present work. Here, the downscaling method was applied to a given cloud type and a robust description of the statistics of the obtained high-resolution relative humidity will be incomplete without also considering other cloudy situations (icy, liquid and mixed phase).

9. Does the scattering ratio contain enough information for all of these relevant geophysical factors?

As demonstrated by Figure 4, 5, and 7 for tropical ice clouds, the scattering ratio measured by CALIPSO, which depends on the clouds microphysical properties, is a good predictor of the relative humidity as measured by SAPHIR. Adding  more predictors like temperature could indeed be interesting, but this would require co-locating a third source of measurements, inevitably reducing the training set.

10. If the authors made plots/summary statistics of RH at different scales and compared to the published literature, that would go a long way towards proving the utility of this approach.

As already mentioned, we agree on this point, but we believe that this would make more sense with a larger sample size and the extension of the method to other cloud types, which is currently work in progress.

11. Also, wouldn't better process-relevant covariances be obtained from a profile by profile matching of SR to SAPHIR?

The statistical method cannot be calibrated with only one profile at a time. It needs a sample of many profiles to learn the relationship between the predictors (here SR) and the predictand (here RH). Hence, the ``profile-by-profile'' approach cannot be implemented.

12. It is not well explained why clustering was used (besides the issues of "mixed phase" brought up in a response to a reviewer, but for clouds colder than 240 K or so there should be very few of these clouds).

The reason of using a statistically-based clustering approach is twofold. First, as explained in section 4.2, the "mixed" flags resulting from the averaging procedure, requires some physical interpretation of these mixed pixels (e.g. are ICE-MIX, ICE-LIQ-MIX profiles representing the same vertical cloud structure?), while a statistically-based clustering method encompasses this problem Additionally, by using the k-means approach, which allows to increase the number of clusters, the method might be better generalizable to boundary layer clouds. The latter are in fact characterized by a much larger variety in the SR vertical structure (c.f. Fig. 2), which leads to more varied profiles (not shown) when using a global cloud flag that does not account for the order of the pixel values.

This being said, the use of k-mean clusters, instead of the cloud phase-based flags, does not change the results of this study as shown in Fig. 7, which shows that the CRPSS scores for the results derived from ICE/ICE-MIX phase flagged profiles in July 2013 in the Indian Ocean are very similar to those obtained in the same month and ocean basin using the profiles selected by the k-mean approach.

We expanded the discussion on the reasons of adopting the k-means approach in the definitions of the SR clusters (page 7, lines 15-21) as well as the description on the method to select the cluster that best represents tropical ice cloud profiles (page 8, lines 4-8). We also added to Fig. 7 the CRPSS results for ICE/ICE-MIX profiles for July 2013 in the Indian Ocean.

**Reviewer 3:**

**Minor comments**

1. # page 30, Figure 9
 # page 13, lines 3-12

The "extrapolation" on the 1-km strip to the side of the CALIPSO track does not seem to make sense. At least the result (lower part of the figure) is unrealistic. Why should a CALIPSO pixel also be representative to the side if the neighbouring pixels show the opposite (strong variation) in the flight direction? It would be useful to express this even more clearly.

Figure 9 suggests that almost all the RH variability is explained by the SR predictors, and any spatial residual effect is negligible. This explains why, as noted by the reviewer, each pixel also seems representative of the pixels in the direction orthogonal to the flight direction (where cloud observations are not available and we had to assume that each SR profile is also representative of the cloud distribution along the direction orthogonal to CALIPSO track within a distance of 1 km), while we can see strong variations along the flight direction. However, this does not imply that there are no variations to the side of each pixel. Instead, what this result shows is that the model is not improved by accounting for any spatial residual random effect.

We have added some text in the manuscript (page 14, line 34 – page 15, line 2 ) to clarify this point.

2. # page 7 line 3
What the "1" means here? "both clusters 1 derived by"

Figure 3 displays the clusters defined after running the k-means method for 8 and 13 clusters. The "1" means "the cluster named 1". We modified this sentence and it now reads "As Fig. 3 shows, both clusters named "1" derived by k-means (…)".

3. # page 12 line 2
"(L6, $R^2 \sim 0.3$)", but in figure 5 I found "$R^2=0.4$"

Indeed, this is a typo and we corrected it.

4. # page 14 line 23
"the next tow decades" >> "two"?

We corrected the typo.

5. # page 23, Figure 2
In the color scheme of upper plot, grey is defined between 0.1 and 1.2. And yellow between 1.2 and 5. In description of figure I can read: "and $1.2 < SR < 5$ (grey)". But nothing to range from 1.2 to 5. Same color scheme is used in figures 1b, 3, 8a.
Please correct it, so that both the color scheme and the description match.

We corrected the legend and the colour scale in the Figure.

6. # page 26, Figure 5s
the color scheme is not optimal: all counts lower than 30, I cannot see at a printed page

We added a contour line to each pixel to make the pixels corresponding to the lower counts more visible.

7. # page 27, Figure 6

label of X-axis is too scarce, because the shown numbers are Altitude (km)

We changed the x-axis label.

**Reviewer 4:**

1. Overall this manuscript presents an interesting analysis. My main concern is I'm not completely sure how to interpret the fidelity of the downscaling without some additional independent validation with high resolution measurements. The authors should be more clear on this point.

The downscaling scheme presented in this study differs from the classical downscaling approach where local variables, generally point-scale observations, are generated from large-scale variables, available at the much coarser grid-scale resolution typical of climate models and reanalyses outputs, and some point-scale covariate(s) at the same fine-scale spatial resolution as the response variable (e.g. elevation data). For this purpose, amongst other methods, also regression-based methods have been used (Vrac et al., 2007), where the model is trained on the available local variables, representing the ground truth. In this case, the evaluation of the fidelity of the downscaling is straightforward, as one can compare the predictions from the model to local observations that were not used for training (e.g., Vaittinada Ayar et al., 2015).

However, in the case presented in this study, no RH observations at the horizontal resolution of the cloud measurements (or higher) are available such that, when co-located with CALIPSO data, provide a large enough training or even testing set for the regression model. This means that in order to obtain some estimates of RH that vary with cloud profiles, we are forced to the opposite approach, where the coarse RH observations measured by SAPHIR are taken as the ground truth, and are regressed against the cloud profiles. Given that the cloud profiles are measured at finer resolution, we refer at the predictions derived in this way as downscaling, since we can incorporate the higher-resolution variability of the covariates in the estimates of the response variable.

In this context, without some additional independent validation with high resolution measurements, the accuracy of the predictions cannot be directly assessed since the model error cannot be quantified at the level of the finer-resolution observations. On the other hand, by adopting the Quantile Random Forest model, we are able to provide uncertainty estimates in the model predictions that account for the RH variability (at the resolution of the coarse scale measurements), while applying the 'mass-balance' correction ensures the best possible consistency with the original measured values.

Clearly no point-to-point validation can be reasonably performed considering the time scales of *in-situ* or ground-based measurements vs. satellite measurements. However, it might still be possible to gain insights on the quality of the downscaling by statistically comparing the RH distributions from available higher-resolution instruments (e.g. water vapor profiles from lidar collected by recent airborne field campaigns) and the downscaled profiles derived with the method presented in this study. Nevertheless, this will require extending the method on all years and locations of available data as well as to other cloud types, which is beyond the scope of the present study.

To make our point clearer, we expanded the discussion on the definition of the term downscaling in the current approach (page 11, line 23 – page 12 line 28).

2. Page 1, Line 12: Should be "will help"

We corrected the typo.

3. Page 2, line 16: I'm not sure what "surface-blind" means in this context.

This means that the sensors provide profiles that are not affected by the surface reflectivity.

4. Page 2, line 31: "clouds" -> "cloud"

We corrected the typo.

5. Page 3, line 9: "backscattered" -> "backscatter"

We replaced it with "attenuated backscatter ratio".

6. Page 3, line 12: delete "to"

We corrected the typo.

7. Page 4, line 1: "spaceborn" -> "spaceborne"

We corrected the typo.

8. Page 4, line 12: "to predict the (downscaled) water-vapour vertical structure using cloud profiles only" This statement is confusing since later in this paragraph the authors state that they are looking horizontally and that the current study is not considering the vertical heterogeneity. Please clarify the wording.

We understand the confusion and corrected the sentence. It now reads: "to predict the (downscaled) RH layered profiles using cloud profiles only."

9. Page 14, line 9: Several studies have looked at biases in reanalysis upper tropospheric humidity and should also be cited here:

Davis, S. M., Hegglin, M. I., Fujiwara, M., Dragani, R., Harada, Y., KOBAYASHI, C., Long, C., Manney, G. L., Nash, E. R., Potter, G. L., Tegtmeier, S., Wang, T., Wargan, K. and Wright, J. S.: Assessment of upper tropospheric and stratospheric water vapor and ozone in reanalyses as part of S-RIP, Atmospheric Chemistry and Physics, 17(20), 12743–12778, doi:10.5194/acp-17-12743-2017, 2017.

Jiang, J. H., Su, H., Zhai, C., Wu, L., Minschwaner, K., Molod, A. M. and Tompkins, A. M.: An assessment of upper troposphere and lower stratosphere water vapor in MERRA, MERRA2, and ECMWF reanalyses using Aura MLS observations, J Geophys Res-Atmos, 120(22), 11,468–11,485, doi:10.1002/2015JD023752, 2015.

We added these references in the text (page 15, line 32).

10. Page 14, line 23: "tow" -> "two"

We corrected the typo.

**Reviewer 5:**

Review of the manuscript "Statistical downscaling of water vapour satellite measurements from profiles of tropical ice clouds" by Giulia Carella, Mathieu Vrac, Hélène Brogniez, Pascal Yiou, and Hélène Chepfer.

In the manuscript, a novel approach for approximating the relative humidity (RH) in the middle atmosphere from the observations of scattering ratio (SR) of the CALIPSO satellite instrument is presented. Several machine learning models are exploited as the primary approach along with additional iterative corrections aimed to preserve the average estimated RH to be close to observed by SAPHIR instrument. Preprocessing of source data includes spatial co-location of SAPHIR and CALIPSO sounding locations; averaging of characteristics of cloud profiles above 2km; filtering of CALIPSO observations related to ICE and ICE-MIX clouds based on k-means clustering results and additional filtering.

Overall I find the idea and the approach of the study to be very promising. The models exploited in the study are described in a clear manner, without over-complications or obscured details. I find the English of the manuscript a bit knotty, but acceptable.

1. General comment: The main concern about the presented study is the unclear general setup of the problem. It is clear that the covariates for a model (one of five regression models) are SR values preprocessed in a way described in Section 2.2. It is also clear that the target values of a model are RH at the points at fine spatial resolution colocated with CALIPSO sounding events. However, it is unclear which values are the ground truth for these target values. Are they just the value of SAPHIR pixel RH? Then it should be mentioned clearly.

This is correct, the mean values of the RH distribution retrieved by SAPHIR are the ground truth. These coarser RH profiles have been evaluated with radiosoundings and thus their uncertainties are documented (Clain et al, JAOT, 2015). As also explained below (see the answer to comment 3), we modified the text (page 11, line 23 – page 12 line 28) to clarify this point.

2. As far as I understand the approach of the authors, the general setup of the problem is as follows: SR (at p=21 height layers) are the covariates for a model. SR are defined at fine spatial (horizontal) resolution. RH$l$ (at 6 pressure levels) are target values for a model. RH$l$ are defined at coarse spatial resolution. A model is fitted to estimate RH$l$ at each point of CALIPSO sounding based on SR values of these points. There is also a step for preserving the so-called "mass balance." And there is no "ground truth" for RH values at the fine spatial resolution other than coarse SAPHIR data.

This is correct and has been clarified in the text (page 11, line 23 – page 12 line 28).

3. Following this setup, the authors fitted the models they presented in Section 3.3.1. As a result, the authors obtained RH at a fine (horizontal) resolution with some variability within each SAPHIR pixel. However, there is no reason to claim this variability to be downscaled structure of RH within SAPHIR pixel. The uncertainty of the RH$l$ value estimated be a model within a SAPHIR pixel is explained by two factors: RH variability itself and errors of a model. The authors did not mention in any way how did they determine the part of the uncertainty that is related to RH variability. And, as far as I understand the setup of the presented study, there is no reason to claim the variability of RH$l$ value estimated by a model to be a fine structure of RH. It is especially arguable in case of non-parametric models like RF or QRF.

The downscaling scheme presented in this study differs from the classical downscaling approach where local variables, generally point-scale observations, are generated from large-scale variables, available at the much coarser grid-scale resolution typical of climate models and reanalyses outputs, and some point-scale covariate(s) at the same fine-scale spatial resolution as the response variable (e.g. elevation

data). For this purpose, amongst other methods, also regression-based methods have been used (e.g., Vrac et al., 2007), where the model is trained on the available local variables, representing the ground truth. In this case, the evaluation of the fidelity of the downscaling is straightforward, as one can compare the predictions from the model to local observations that were not used for training (e.g., Vaittinada Ayar et al., 2015).

However, in the case presented in this study, no RH observations at the horizontal resolution of the cloud measurements (or higher) are available such that, when co-located with CALIPSO data, provide a large enough training or even testing set for the regression model. This means that in order to obtain some estimates of RH that vary with cloud profiles, we are forced to the opposite approach, where the coarse RH observations measured by SAPHIR are taken as the ground truth, and are regressed against the cloud profiles. Given that the cloud profiles are measured at finer resolution, we refer at the predictions derived in this way as downscaling, since we can incorporate the higher-resolution variability of the covariates in the estimates of the response variable.

In this context, without some additional independent validation with high-resolution measurements, the accuracy of the predictions cannot be directly assessed since the model error cannot be quantified at the level of the finer-resolution observations. On the other hand by adopting the Quantile Random Forest model, we are able to provide uncertainty estimates in the model predictions that account for the RH variability (at the resolution of the coarse scale measurements), while applying the 'mass-balance' correction ensures the best possible consistency with the original measured values.

Clearly, no point-to-point validation can be reasonably performed considering the time scales of *in-situ* or ground-based measurements vs. satellite measurements. However, it might still be possible to gain insights on the quality of the downscaling by statistically comparing the RH distributions from available higher-resolution instruments (e.g. water vapor profiles from lidar collected by recent airborne field campaigns) and the downscaled profiles derived with the method presented in this study. Nevertheless, this will require extending the method on all years and locations of available data as well as to other cloud types, which is beyond the scope of the present study.

The fact that within the framework presented in this study, at the finer resolution scale, the model error cannot be directly separated from the variability in the response variable, might create some confusion on the meaning of the term "downscaling" as adopted here. Nonetheless, for the model estimates, the variance explained by the cloud profiles is, by construction, higher than that for SAPHIR measurements, and this serves as a justification for the downscaling term: the predictions from the model are better correlated with the higher-resolution cloud profiles, and can therefore be considered as a downscaled product, in the sense discussed above.

To make our point clearer, we expanded the discussion on the definition of the term downscaling in the current approach (page 11, line 23 – page 12 line 28).

4. According to the above, the naming of the study is confusing and misleading. I would totally agree with the title mentioning "approximation" or "regression" of RH based on CALIPSO profiles. However, the "downscaling" naming seems incorrect in this case.

As explained in the answer to the previous comment, we think that the term "downscaling" is rightly applicable in this case. Therefore, we kept the current title.

5. Considering those mentioned above, the discussion about the downscaling results is also unconvincing (p.12 L27 - p.13 L12). Taking the design of the study into account, one cannot claim to be able to obtain variations of RH within SAPHIR pixels. The conclusions related to the downscaling feature of the presented approach are incorrect in this case as well. The results of a model can be considered as an estimate of RH with some uncertainty, but cannot be considered as a downscaling product with estimates of inhomogeneities.

As explained in the answer to comment 3, we think that the term downscaling is well justified, and it is fair to claim that we obtain variations of RH within each SAPHIR pixel that are correlated with the cloud profiles.

6. Section 3.2: The classification of the SR profiles based on clustering techniques does not seem reliable. Since clustering itself is a tool for explorative analysis rather than classification, one cannot claim the clustering result to be the answer for the question "which profiles correspond to ice clouds and which do not." It is not clear how do authors checked if the profiles of the cluster 1 indeed correspond to ICE and ICE-MIX observations (p. 7, L3-9). As it seems at the current state of the manuscript, the similarity of clustering made with k-means with the observations characterized as ICE and ICE-MIX was assessed visually. If that is the case, it does not seem like a convincing way of detection of ice clouds. It is also not clear why the authors did not use cloud phase flags instead of this additional clustering approach.

The reason of using a statistically-based clustering approach is twofold. First, as explained in section 4.2, the "mixed" flags resulting from the averaging procedure, requires some physical interpretation of these mixed pixels (e.g. are ICE-MIX, ICE-LIQ-MIX profiles representing the same vertical cloud structure?), while a statistically-based clustering method encompasses this problem Additionally, by using the k-means approach, which allows to increase the number of clusters, the method might be better generalizable to boundary layer clouds. The latter are in fact characterized by a much larger variety in the SR vertical structure (c.f. Fig. 2), which leads to more varied profiles (not shown) when using a global cloud flag that does not account for the order of the pixel values.

Concerning the attribution of the k-means clusters to ICE/ICE-MIX profiles, the cluster whose mean SR profile visually appears the most "similar" is also that with the smallest distance from the mean SR profile classified by the ICE/CE-MIX phase flag. The distance was computed as the weighted Euclidean distance between each pixel of the mean SR k-mean-derived profile and the corresponding pixel in the mean ICE/ICE-MIX SR profile, with weights defined by the presence/absence of clouds (we used unitary weights if both pixels where cloudy (SR>5) and a weight of 9999 otherwise).

This being said, the use of k-mean clusters instead of the cloud phase-based flags, does not change the results of this study as shown in Fig. 7, which shows how the CRPSS scores for the results derived from ICE/ICE-MIX phase flagged profiles in July 2013 in the Indian Ocean are very similar to those obtained in the same month and ocean basin using the profiles selected by the k-mean/weighted distance approach.

We expanded the discussion on the reasons of adopting the k-means approach in the definitions of the SR clusters (page 7, lines 15-21) as well as the description on the method to select the cluster that best represents tropical ice cloud profiles (page 8, lines 4-8). We also added to Fig. 7 the CRPSS results for ICE/ICE-MIX profiles for July 2013 in the Indian Ocean.

7. Section 4: The results described in Section 4 (p. 12, L1-2) are inconsistent with Figure 5 in terms of the values of the coefficient of determination. $R^2$ for L6 mentioned to be ~0.3 in the text of the manuscript; meanwhile, it is 0.4 in the Fig.5. Also, for the layers L1-L5 the values of $R^2$ claimed to be >=0.7, and as presented in the figure, are precisely 0.7. While formally, it is not a mistake; it seems like a mistyping in the figure.

We corrected the typo in the figure.

8. Fig.7: The description of results shown in Fig.7 is not clear. It is unclear what is the statement to which "Similar results can be found for different choices of the number of clusters…" is related (p.12, L12-15).

We rewrote this description. It now reads:

"Finally, as Fig 7 shows, the CRPSS distribution is similar for different choices of clusters (k-means with k=8 and k=13 and for the cluster corresponding to profiles with ice cloud pixels only) as well as for different seasons (July and January) and regions (Indian Ocean and Pacific Ocean): for all the layers, the median CRPSS is positive, which confirms the robustness of the approach."

9. Fig.3: The horizontal axis meaning is not clear enough. It should be labeled with some kind of "cluster number" of another informative label.

We modified the x-axis labels of Fig 3 as" cluster name".

[revised manuscript text omitted]

---

## Author Response (AR4)

**Reviewer 1:**

Suggestions for revision or reasons for rejection (will be published if the paper is accepted for final publication)

The authors did a pretty good job in answering my comments from the last iteration. The additional chapter "Physical approach and related limitations" describes very nicely the caveats of the approach. So, even if I might have still my doubts about the product, I cannot say that the authors are not honest about it.
I assume the data will get released sooner or later anyway. Then I would rather have this paper published, to make sure that users understand what they are dealing with.

So, I would propose to consider this article for publication.

I have just a few technical comments.

We would like first to thank the Reviewer for his feedback. We have modified the text according to all his suggestions.

Abstract:
Add one more sentence after the second sentence (I copied it from chapter 3):
"Here, we present a method to downscale observations of relative humidity available from the passive microwave sounder SAPHIR at a nominal horizontal resolution of 10 km to the finer resolution of 90 m using scattering ratio profiles from the lidar CALIPSO. The small scale water vapour is hereby deduced from the indirect physical correlation between RH and the lidar observations."

Cut the next sentence in two pieces to make it more readable:
"With the scattering ratio 5 profiles as covariates, an iterative approach applied to a non-parametric regression model based on Quantile Random Forest is used. It allows to effectively incorporate the high-resolution variability from cloud profiles into the predicted relative humidity structure"

We modified the abstract and now reads:

Multi-scale interactions between the main players of the atmospheric water cycle are poorly understood, even in present-day climate, and represent one of the main sources of uncertainty among future climate projections. Here, we present a method to downscale observations of relative humidity available from the passive microwave sounder SAPHIR at a nominal horizontal resolution of 10 km to the finer resolution of 90 m using scattering ratio profiles from the lidar CALIPSO. With the scattering ratio profiles as covariates, an iterative approach applied to a non-parametric regression model based on Quantile Random Forest is used. This allows to effectively incorporate into the predicted relative humidity structure the high-resolution variability from cloud profiles. The finer-scale water vapour structure is hereby deduced from

the indirect physical correlation between relative humidity and the lidar observations. Results are presented for tropical ice clouds over the ocean: based on the coefficient of determination (with respect to the observed relative humidity) and the Continuous Rank Probability Skill Score (with respect to the climatology), we conclude that we are able to successfully predict, at the resolution of cloud measurements, the relative humidity along the whole troposphere, yet ensuring the best possible coherence with the values observed by SAPHIR. By providing a method to generate pseudo-observations of relative humidity (at high spatial resolution) from simultaneous co-located cloud profiles, this work will help revisiting some of the current key barriers in atmospheric science. A sample dataset of simultaneous co-located scattering ratio profiles of tropical ice clouds and observations of relative humidity downscaled at the resolution of cloud measurements is available at http://dx.doi.org/10.14768/20181022001.1 (Carella et al., 2019).

Chapter 3:
"This work is rooted on the following physics"
Replace it by something like "We use the following assumption in our approach"

This sentence now reads:
"This work is based on the following physical assumption."

Same chapter, page 7, line 6:
There is a "d" too much : Correct to "To avoid the misuse"

Corrected.

Same page, further down - line 9:
I would replace "relevant" with "applicable"

Changed.

[revised manuscript text omitted]